## Learning to filter: Snow data assimilation using a Long Short-Term Memory network

Giulia Blandini<sup>1,2</sup>, Francesco Avanzi<sup>1</sup>, Lorenzo Campo<sup>1</sup>, Simone Gabellani<sup>1</sup>, Kristoffer Aalstad<sup>3</sup>, Manuela Girotto<sup>4</sup>, Satoru Yamaguchi<sup>5</sup>, Hiroyuki Hirashima<sup>5</sup>, and Luca Ferraris<sup>1,2</sup>

#### Abstract.

10

Trustworthy estimates of snow water equivalent and snow depth are essential for water resource management in snow-dominated regions. While ensemble-based data assimilation techniques, such as the Ensemble Kalman Filter (EnKF), are commonly used in this context to combine model predictions with observations therefore to improve model performance, these ensemble methods are computationally demanding and thus face significant challenges when integrated into time-sensitive operational workflows. To address this challenge, we present a novel approach for data assimilation in snow hydrology by utilizing Long Short-Term Memory (LSTM) networks. By leveraging data from 7 diverse study sites across the world to train the algorithm on the output of an EnKF, the proposed framework aims to further unlock the use of data assimilation in snow hydrology by balancing computational efficiency and complexity.

We found that a LSTM-based data assimilation framework achieves comparable performance to state estimation based on an EnKF in improving open-loop estimates with only a small performance drop in terms of RMSE for snow water equivalent (+ 6 mm on average) and snow depth (+ 6 cm), respectively. All but 2 out of 14 site-specific-LSTM configurations improved on the Open Loop estimates. The inclusion of a memory component further enhanced LSTM stability and performance, particularly in situations of data sparsity. When trained on long datasets (25 years), this LSTM data assimilation approach also showed promising spatial transferability, with less than a 20% reduction in accuracy for snow water equivalent and snow depth estimation.

Once trained, the framework is computationally efficient, achieving a 70% reduction in computational time compared to a parallelized EnKF. Training this new data assimilation approach on data from multiple sites showed that its performance is robust across various climate regimes, during dry and average water-year types, with only a limited drop in performance compared to the EnKF (+6 mm RMSE for SWE and +18 cm RMSE for snow depth). This work paves the way for the use of deep learning for data assimilation in snow hydrology and provides novel insights into efficient, scalable, and less computationally demanding modeling framework for operational applications.

<sup>&</sup>lt;sup>1</sup>CIMA Research Foundation, Savona, Italy

<sup>&</sup>lt;sup>2</sup>DIBRIS, University of Genoa, Genova, Italy

<sup>&</sup>lt;sup>3</sup> Department of Geosciences, University of Oslo, Oslo, Norway

<sup>&</sup>lt;sup>4</sup>Department of Environmental Science, Policy, and Management, University of California, Berkeley, Berkeley, CA, United States of America

<sup>&</sup>lt;sup>5</sup> Snow and Ice Research Center, National Research Institute for Earth Science and Disaster Resilience, Nagaoka, Japan **Correspondence:** Giulia Blandini (giulia.blandini@edu.unige.it)

#### 1 Introduction

When studying the hydrological cycle, one cannot underestimate the key role played by snow (Pagano and Sorooshian, 2002); indeed, for snow-dominated catchments, today's snow is tomorrow's water. Information on the state and distribution of snow cover provides helpful information to characterize seasonal water storage (Zakeri et al., 2024), seasonal to annual water availability (Metref et al., 2023), and several cascading socio-hydrologic implications (Avanzi et al., 2024).

Especially in cold regions, which are heavily affected by climate change (Hock et al., 2019), the snowpack often functions as the primary source of streamflow, particularly during spring and summer (Bales et al., 2006). Moreover, considering the high spatial variability in these regions, the scientific community agrees on the needs of reliable estimates of Snow Water Equivalent (SWE) and snow depth in snow-dominated environments, which are essential for effective and timely management of water resources (Hartman et al., 1995).

However, the models used in operational snow hydrology are hampered by uncertainties (Beven, 2012). Uncertainties arise from the accuracy and reliability of the equations and their discretization used to numerically represent physical processes on a computer (structural uncertainty), as well as from model inputs (e.g., meteorological uncertainty) and model parameters (parametric uncertainty, see Girotto et al., 2020). To constrain this uncertainty, independent snow-related data sources such as ground-based measurements or remotely sensed measurements can be used (Tsai et al., 2019), but all observations are also subject to inherent uncertainty in the form of unknown observation and representation errors (Gascoin et al., 2024; Van Leeuwen, 2015). Ground-based snow measurements, for example, are limited to environmental conditions at the point-scale, which are often influenced by instrumental noise as well as local distortions by wind, topography, and vegetation, which pose challenges at the scale of a model grid cell (Malek et al., 2017). It is also worth mentioning that their representativeness is expected to degrade in the future due to evolving climate and land surface conditions, further limiting their utility for large-scale modelling efforts (Cowherd et al., 2024b). In contrast, remote sensing provides spatially explicit information, but its measurements are frequently constrained by a coarse spatial resolution and additional uncertainties in retrieval algorithms (Aalstad et al., 2020).

Given the uncertainties inherent in both models and measurements, data assimilation presents a promising framework to optimally combine them (Evensen et al., 2022), so as to provide a statistically optimal estimate of the snowpack state. In the recent decade, snow data assimilation has progressed from a limited number of case studies to more established and widely used techniques (Largeron et al., 2020; Girotto et al., 2020; Alonso-González et al., 2022), largely driven by advancements in satellite data products and computational resources (Houser et al., 2012; Aalstad et al., 2018; Deschamps-Berger et al., 2023; Lievens et al., 2022; Mazzolini et al., 2024). Commonly assimilated variables include snow-covered area (SCA) (Margulis et al., 2016), snow depth (Girotto et al., 2024) and SWE (Magnusson et al., 2014). Recent research has also begun to explore the potential of thermal infrared sensors and radar data (Alonso-González et al., 2023; Cluzet et al., 2024). From a methodological point of view, while traditional methods such as direct insertion or nudging (Boni et al., 2010) are still widely used, research interest in this field is increasingly shifting towards Bayesian data assimilation techniques such as the Ensemble Kalman Filter (EnKF)

and the Particle Filter (PF) (Evensen et al., 2022). These Bayesian methods, which account for uncertainties both in the model and the observations, have demonstrated significant improvements in predictions of snow and streamflow variables (Huang et al., 2017; Alonso-González et al., 2022; Metref et al., 2023). On the other hand, they typically incur high computational costs (Girotto et al., 2020), which are often incompatible with operational procedures (Pagano et al., 2014).

60

The high computational cost of ensemble-based (Monte Carlo) data-assimilation techniques such as the EnKF and the PF arises from the need to perform a large ensemble (collection) of model predictions, which can strain computational capacity and extend processing times (Evensen et al., 2022). Consequently, the deployment of these ensemble-based techniques in real-time applications can be challenging, necessitating efficient algorithms and robust computing infrastructure to ensure timely and accurate results. Decreasing such computational requirements would allow one to obtain estimates with a shorter turnaround and/or to increase model complexity with the same computational burden. In this context, it is worth mentioning the work of Oberrauch et al. (2024), one of the few studies that addresses the challenge of implementing a particle-based data assimilation scheme for large-scale, fully distributed, near real-time snow modelling applications, effectively balancing computational feasibility with operational efficiency.

Recently, Deep Learning (DL) has gained attention for its ability to model complex system dynamics without requiring detailed knowledge of physical processes or relying on strict structural assumptions (Sit et al., 2020). Based on interconnected neural networks, these model architectures excel at learning system dynamics from large datasets, and may overcome the structural limitations that challenge traditional physically-based models (LeCun et al., 2015; Murphy, 2023). Among the most commonly used Deep Learning architectures, Long Short-Term Memory (LSTM) networks (Hochreiter and Schmidhuber, 1997), a type of recurrent neural networks, can memorize internal system states and capture long-term dependencies between inputs and outputs. LSTM networks have demonstrated significant success in predicting time-series data, particularly in hydrological applications, where they have shown comparable performance to traditional physically-based models (Fan et al., 2020; Chen et al., 2023; Kratzert et al., 2018, 2019). Due to the strong temporal autocorrelation and memory of the snowpack (Fiddes et al., 2019), these networks appear to be especially well suited for snow analysis.

In the broader field of operational hydrology, Boucher et al. (2020) pioneered a novel ensemble-based data assimilation approach leveraging neural networks. However, the use of Deep Learning for data assimilation remains largely underexplored in the field of snow hydrology. One exception is the recent study by Guidicelli et al. (2024), who combined ensemble-based data assimilation with Deep Learning to improve spatio-temporal estimates of SWE using sparse ground track data in the eastern Swiss Alps. This approach utilized an Iterative Ensemble Smoother, an iterative batch-smoother variant of the EnKF, in conjunction with a degree-day model to reconstruct SWE temporal evolution, while a feedforward neural network (FNN) facilitated spatial propagation based on topographic features. As a more recent exception of combining Deep Learning and snow data assimilation, Song et al. (2024) developed an LSTM-based framework to assimilate lagged observations of SWE or satellite-derived snow cover fraction (SCF) over the western U.S., aiming to improve seasonal snow predictions. While their approach further consolidates the potential of Deep Learning for data assimilation in snow hydrology, it relied on a relatively simple assimilation setup, dealing with long lagged time step rather than a consequential and quasi real time approach. Other than these initial attempts, and the body of work on stand-alone Deep Learning for snow modelling (Cui et al., 2023; Daudt

et al., 2023), the potential of combining advanced Deep Learning and data assimilation algorithms for predicting snowpack dynamics remains largely unexplored.

Building on the concept of Deep Data Assimilation introduced by Casas et al. (2020) and Arcucci et al. (2021) as well as a growing literature of related methods (Cheng et al., 2023), this research aims to enhance data assimilation methods in snow hydrology by proposing an alternative approach for assimilating snow-related quantities, specifically SWE and snow depth, through the use of LSTM networks. These networks will be trained on the output from an EnKF, with the goal of improving snowpack estimations while minimizing computational efforts. Here we utilize S3M, a hybrid temperature-radiation-driven cryospheric model (Avanzi et al., 2022), as our dynamical model combined with the state analysis (updates) of an EnKF to train an LSTM to assimilate SWE and snow depth data in S3M for 7 disparate study sites across the northern hemisphere. The study will focus on investigating four main research questions: (i) What is the performance of a LSTM network in filtering, especially in comparison with an EnKF? (ii) How does the performance of the network respond to data sparsity? (iii) Is it feasible to transfer an LSTM algorithm trained on one site to other sites without a significant loss in performance? (iv) How does the performance of the model vary between different types of water years?

#### 2 Materials and Method

#### 105 **2.1 Data**

110

115

When working with Deep Learning algorithms, the quality of the dataset is crucial, as the performance of the trained network will highly depend on it (He et al., 2019). Hence, in this study we employed high-quality, pre-processed datasets from long-term, internationally acknowledged snow research stations across the northern hemisphere (Figure 1). The datasets used where those of precipitation (mm/h), solar radiation (W/m $^2$ ), relative humidity (%), air temperature ( $^{\circ}$ C), and daily average temperature ( $^{\circ}$ C) along with SWE (mm/h) and snow depth (cm) ground measurements.

Here is a list of the station locations, along with their associated reference papers and abbreviations:

- Torgnon, Aosta Valley, Italy TRG (Filippa et al., 2015).
- Col De Porte, Isère, France CDP (Lejeune et al., 2019).
- Weissfluhjoch, Davos, Switzerland WFJ (Wever, 2017).
- Kühtai, Tirol, Austria KHT (Krajči et al., 2017).
  - FMI-ARC Sodankylä Geophysical Observatory, Finnish Lapland FMI-ARC (Essery et al., 2016).
  - Nagaoka, Japanel NGK (Avanzi et al., 2019).
  - Reynolds Mountain East, Idaho, USA RME (Reba et al., 2011).

The sites were selected to ensure geographic and climatic diversity, spanning various regions that are exposed to a variety of snow climates (Sturm and Liston, 2021), (see Table 1 and Table 2). The characteristics of the site vary widely, with elevations ranging from lowland areas such as Sodankylä (179 m) to high alpine environments such as Weissfluhjoch (2540 m). Annual and winter precipitation varies significantly across different locations, ranging from relatively dry areas like Torgnon, with an annual average of 794 mm, to much wetter regions such as Nagaoka, which receives 2773 mm per year. For this comparative analysis, winter is defined as the meteorological winter in the northern Hemisphere, spanning the months of December through February. Air temperature ranges reflect this environmental diversity, encompassing cold alpine regions, temperate meadows, and wetlands.

| Site    | Description               | Altitude (m a.s.l) | MAP (mm) | MWP (mm) | MAAT [min,max] (°C) |
|---------|---------------------------|--------------------|----------|----------|---------------------|
| TRG     | Subalpine grassland       | 2160               | 794      | 161      | 3 [-15, 20]         |
| CDP     | Grassy meadow             | 1325               | 1896     | 550      | 6 [-13, 17]         |
| WFJ     | Almost flat area          | 2540               | 1631     | 391      | -1 [-21,17]         |
| KHT     | Steep alpine valley       | 1920               | 1131     | 186      | 3 [-18, 22]         |
| FMI-ARC | Large wetland area        | 179                | 551      | 125      | 0 [-35, 27]         |
| NGK     | Flat meadow               | 97                 | 2773     | 1104     | 12[-5,36]           |
| RME     | Unsheltered mountain area | 2137               | 817      | 350      | 5 [-20, 30]         |

**Table 1.** Geographic and climatic characteristics (annual precipitation and air temperature statistics) at the selected study sites. MAP=mean annual precipitation (mm), MWP= mean winter precipitation (mm), MAAT = mean annual average temperature (°C)

| Site    | Peak SWE (mm) | Peak Snow depth (cm) | Snow cover duration               | Snow Type      |  |
|---------|---------------|----------------------|-----------------------------------|----------------|--|
| TRG     | 312           | 11                   | From October to May               | Tundra         |  |
| CDP     | 414           | 14                   | From November to May              | Maritime       |  |
| WFJ     | 802           | 23                   | From October/November to August   | Tundra         |  |
| KHT     | 347           | 15                   | From October/November to mid June | Tundra         |  |
| FMI-ARC | 197           | 8                    | From October to May               | Boreal Forest  |  |
| NGK     | 381           | 14                   | From Novemeber to April           | Maritime       |  |
| RME     | 529           | 17                   | From October to May               | Montane Forest |  |

Table 2. Summary of snow characteristics at the selected study sites. Snow classification by Sturm and Liston (2021).

The record period for each dataset varied depending on the timeframes available at each site. To ensure uniform application of the algorithm, all datasets were resampled to a 1-hour frequency using linear interpolation. This hourly resolution resolves day-night cycles of melting and refreezing, revealing air temperature fluctuations and their relationship with snowpack outflow. In addition, it enables the evaluation of the precipitation dynamics, the primary mass input to the seasonal snowpack (Avanzi et al., 2014).

Measurement errors used in the data assimilation process (see Section 2.3) were assigned according to the specific instrumentation utilized at each site, drawing from a combination of expert knowledge and relevant literature (see Tab. A1 in the Appendix).

Based on data sparsity—defined as the presence of 80% or more of the record period containing missing data—or a low temporal data granularity (i.e., temporal frequency coarser than 1 hour), the datasets were categorized into two groups:

- Low data sparsity: NGK, KTH, FMI-ARC, and RME datasets.
- High data sparsity: CDP, TRG, and WFJ datasets.

#### 2.2 The Model: S3M 1D

"Snow Multidata Mapping and Modelling (S3M)" is a spatially distributed cryospheric model developed to compute the snow mass balance and estimate snowmelt using a combined temperature index and radiation-driven melt approach (Avanzi et al., 2022). S3M also includes processes such as snow settling, liquid water outflow, changes in snow albedo, and the partitioning of precipitation phases. S3M is the cornerstone of several operational chains managed by CIMA Research Foundation, which provide real-time, spatially explicit estimates of snow cover patterns (Avanzi et al., 2023).

For this pilot application of a new deep data assimilation scheme, a point-scale version of S3M has been employed. This version retains all the features of the original S3M model, such as precipitation-phase partitioning, snow mass balance, snow metamorphism, and hydraulics, but it models snow dynamics at one point rather than in grid cells distributed across the landscape.

## 2.3 Ensemble Kalman Filter Assimilation Scheme

Aiming at mimicking an established ensemble-based data assimilation algorithm with a Deep-Learning-based approach, we chose a supervised learning approach to our problem (Murphy, 2022). Hence, the training data had to be derived from the state analysis output by such data assimilation scheme. The assimilation algorithm used as training was designed to focus on retrieving an accurate analysis of the state vector ( $\mathbf{x} \in \mathbb{R}^n$  with n the number of states), including both the wet and dry components of SWE, the density of dry snow (kg/m³), and the snow albedo (-). Given the nonlinear nature of S3M, it was decided to use an ensemble method that approximates the posterior probability density function of the analysis using the mean and covariance matrix (Carrassi et al., 2018; Evensen et al., 2022). Given the high robustness even with a relatively small ensemble (Aalstad et al., 2018), an EnKF scheme was developed in S3M.

Kalman Filters, which are sequential data assimilation techniques, optimally combine linear model simulations and observations based on their respective Gaussian error covariances (Särkkä and Svensson, 2023). The analysis state is obtained by applying a correction to the model forecast (or prior) state, weighted by the Kalman Gain, which incorporates information from both model and observation error covariance.

Mathematically, the Kalman filter cycles between a prediction step, known as the forecast or the prior in DA, propagating the state from the pervious time  $t_{k-1}$  to the current time  $t_k$  using a dynamical model and a subsequent update step, known

## **STUDY SITES**

**Figure 1.** Geographical distribution of study sites used for snow modeling and data assimilation: (left) Reynolds Mountain East (RME) in the United States, (center) European sites including Col De Porte, Isère, France, Weissfluhjoch, Davos, Switzerland (WFJ), Torgnon, Aosta Valley, Italy (TRG), Kühtai, Tirol, Austria (KHT) FMI-ARC Sodankylä Geophysical Observatory, Finnish Lapland (FMI-ARC) in Finland, and (right) Nagaoka (NGK) Japan. Map created using the Free and Open Source QGIS.

as *the analysis* or *the posterior* in DA, where the state is updated by assimilating observations through (Evensen et al., 2022; Särkkä and Svensson, 2023):

$$\boldsymbol{x}_{k}^{a} = \boldsymbol{x}_{k}^{f} + \mathbf{K}_{k} \left( \boldsymbol{y}_{k} - \mathbf{H}_{k} \boldsymbol{x}_{k}^{f} \right), \tag{1}$$

where:

•  $x_k^f \in \mathbb{R}^n$  is the forecast (prior, background) mean model state vector at time  $t_k$ .

- $x_k^a \in \mathbb{R}^n$  is the analysis (posterior, updated) mean model state vector at time  $t_k$ .
- $y_k \in \mathbb{R}^m$  is the vector of the observations at time  $t_k$  where the number of observations  $m \ge 0$  may vary in time.
  - $\mathbf{H}_k \in \mathbb{R}^{m \times n}$  is a linear observation operator, that maps from the model state space to the observation space. The time index of this operator is a reminder that the number of observations m may vary over time.
  - $\mathbf{K}_k$  is the Kalman gain at time  $t_k$ , defined as :

200

$$\mathbf{K}_{k} = \mathbf{P}_{k}^{f} \mathbf{H}_{k}^{T} \left( \mathbf{H} \mathbf{P}_{k}^{f} \mathbf{H}_{k}^{T} + \mathbf{R}_{k} \right)^{-1}$$
(2)

• In the dynamic Kalman gain (2),  $\mathbf{P}_k^f \in \mathbb{R}^{n \times n}$  is the forecast error covariance matrix and  $\mathbf{R}_k \in \mathbb{R}^{m \times m}$  is the observation error covariance matrix while  $^{\mathrm{T}}$  denotes the matrix transpose.

The Kalman gain  $\mathbf{K}_k$  in Equation (2) acts as a weighting factor, balancing the correction term (the innovation) by accounting for the relative uncertainties in the forecasted model state through the forecast error covariance matrix  $\mathbf{P}_k^f$  and in the observations through the observation covariance matrix  $\mathbf{R}_k$ .

Although classical Kalman filters are still widely used in signal processing and related fields (Särkkä and Svensson, 2023), they require both linear and Gaussian models. Despite being based on the same linear Gaussian assumptions as the Kalman filter, many nonlinear extensions of the Kalman filter are able to overcome the strict requirement of a linear model. Among these extensions, the EnKF is particularly well suited for high-dimensional nonlinear geoscientific models by relying on an ensemble of simulations to estimate the prior mean and covariance in the update step (1) (Carrassi et al., 2018; Evensen et al., 2022). In particular, together with particle methods, ensemble Kalman methods make up the ensemble-based methods that are among the current state-of-the art methods for snow data assimilation (Aalstad et al., 2018; Alonso-González et al., 2022).

In the present study, a joint data assimilation scheme was developed to update the system state by jointly assimilating ground-based measurements of snow depth and SWE. Despite albedo being another potential data stream to assimilate (Navari et al., 2018), due to the lack of measurements across the 7 sites, assimilation of albedo measurements was not considered herein. Nonetheless, albedo was updated indirectly, based on the assimilation of SWE and snow depth. Moreover, as the model is a point-based simulation, we could not pursue fractional snow-covered area assimilation and since the EnKF can not handle binary observations binary snow cover was not an option either. Finally, since S3M does not solve the full energy balance or simulate snow-temperature profiles, no surface temperature proxy was assimilated.

Ensemble generation was performed by perturbing meteorological model forcing, which included total precipitation (mm/h), solar radiation (W/m<sup>2</sup>), relative humidity (%), air temperature (°C), and daily average temperature (°C).

To each meteorological forcing data point, an ensemble of multivariate errors was added. These errors were generated as realizations of a multivariate stochastic process designed to have a specified covariance matrix derived from the Gaussianized historical meteorological series. The objective was to produce a multivariate time series of meteorological values in a Gaussian space, ensuring that the imposed covariance matrix matched C, the temporal covariance matrix of the historical observations.

The procedure for constructing the stochastic process is base on similar approaches implemented by Reichle et al. (2007), Lannoy et al. (2010) and Durand and Margulis (2006) and is described below:

## 1. Generation of a random covariance matrix:

A random covariance matrix,  $C_o$ , was generated.

## 2. Construction of a Gaussian stochastic process:

(a) A Cholesky decomposition was performed on  $C_o$ , yielding:

$$\mathbf{L}_o = \text{Cholesky}(\mathbf{C}_o).$$

(b) The multivariate stochastic process was defined as:

$$u_{k+1} = u_k + \mathbf{L}_o \epsilon$$
,

where  $\epsilon \sim \mathcal{N}(0, 0.1)$  represents independent standard normal variables.

# Calculation of the covariance matrix: A realization of the stochastic process was generated, and its covariance matrix, We was computed.

## 4. Imposition of the target covariance matrix:

(a) The Cholesky decomposition of the target covariance matrix, C, was computed:

$$\mathbf{L} = \text{Cholesky}(\mathbf{C}).$$

(b) The stochastic process was constructed to impose the covariance matrix C as follows:

$$\tilde{\boldsymbol{u}}_{k+1} = \tilde{\boldsymbol{u}}_k + \mathbf{L}_o \cdot \boldsymbol{\epsilon}$$

Initially, this process was characterized by the covariance matrix  $\tilde{\mathbf{C}}$ . To transform it into a process with the covariance matrix  $\mathbf{C}$ , the following steps were taken:

(a) The transformation:

$$\tilde{\mathbf{L}} \cdot \tilde{\boldsymbol{u}}_{k+1}$$


where  $\tilde{\mathbf{L}} = \text{Cholesky}(\tilde{\mathbf{C}})$ , was applied, normalizing the covariance matrix  $\tilde{\mathbf{C}}$  to the identity matrix  $\mathbf{I}$ .

(b) A second transformation was applied:

$$oldsymbol{u}_{k+1} = \mathbf{L} \cdot \left( ilde{\mathbf{L}} \cdot ilde{oldsymbol{u}}_{k+1} 
ight)$$

which transformed the identity matrix  ${\bf I}$  into the target covariance matrix  ${\bf C}$ .

5. **Perturbation Calculation:** Finally, to compute the perturbations to be added to the meteorological values, the following expression was used:

$$\Delta = \tilde{\boldsymbol{u}}_{k+1} - \operatorname{mean}(\tilde{\boldsymbol{u}}_{k+1})$$




where mean( $\tilde{u}_{k+1}$ ) represents the ensemble mean.

The use of a stochastic process to generate the ensemble of errors was pivotal to ensure temporal coherence in meteorological perturbations. This procedure was location-specific, tailored to the 7 study sites. The ensemble size was defined as 100 members. It was determined to be suitable for an EnKF, based on literature (Aalstad et al., 2018) and testing.

To improve filter performance and stability, the forecast model state vector  $\mathbf{x}_k^f$  at each time step  $t_k$  was also perturbed. To obtain the perturbation, a series of multivariate Gaussian random error with imposed process noise covariance matrix  $\mathbf{Q}$  was added to each forecast model state vector point. The matrix was retrieved from S3M open loop forecast over the entire historical period for each site. Different versions of the observation operator  $\mathbf{H}_k$  were constructed to allow assimilation with only one observed variable when necessary. Post-processing was applied to the filter outputs to ensure physical consistency, adjusting corrections to the filter output while maintaining the physical relationships between the elements of the state vector. This included constraining the values within a physical range and modulating them accordingly.

## 2.4 Long-short term memory neural network

The development of data assimilation using neural networks was framed as a time series forecasting task, leading to the use of Recurrent Neural Networks (RNNs). RNNs leverage internal memory to process sequences of data, making them useful for time-dependent analysis. However, they often struggle with long-term dependencies due to vanishing or exploding gradients (Tsantekidis et al., 2022). To address this, LSTM networks introduce gate mechanisms (input, forget, output) to control information flow, effectively managing long-term dependencies (Hochreiter and Schmidhuber, 1997).

#### 245 **2.4.1 Data pre-processing**

Effective data pre-processing is critical for the successful application of LSTM networks, as it improves prediction accuracy, reduces computational costs, and enhances model robustness and repeatability (Isik et al., 2012). Proper pre-processing not only accelerates network convergence, but also helps the model capture essential patterns in the data. For LSTM networks, which are sensitive to the distribution and scale of the inputs, pre-processing plays a key role in mitigating issues like exploding or vanishing gradients and managing differences in feature magnitudes.

Data pre-processing in this study involved two key steps:

• **Distribution adjustment:** Snow related variables frequently hit the lower physical boundary of 0 mm of SWE or 0 cm of snow depth, posing challenges for the LSTM, which struggled to handle this behavior. To overcome this, the data range was extended by redefining the lower limit to a value below zero.

• Scaling with historical data: After adjusting the distribution, the input values were also standardized using the mean and standard deviation calculated from historical records at each site. This standardization ensured that all input features were on a consistent scale.

## 2.4.2 Custom loss function




To ensure compliancy of the LSTM predictions to specific problem domain constraints, a custom loss function was developed.

This loss function comprises two main components:

- Root Mean Square Error (RMSE): This measures the difference between the LSTM predictions  $x_k^{a\star}$  and the analysis state vectors generated by the EnKF,  $x_k^a$ . By minimizing RMSE, the model was trained to closely follow the reference trajectory provided by the EnKF.
- Physics-based Regularization Term: An additional U-shaped penalty function was introduced to enforce adherence to physical constraints and guide the model towards specific physical behaviors. This term penalizes the network for making predictions that violate predefined physical boundaries. The function is expressed as:

$$\operatorname{Loss}(\boldsymbol{x}_{k}^{a\star}) = \frac{1}{\prod_{i=1}^{n} \left| \boldsymbol{x}_{k,i}^{a\star} - \boldsymbol{a}_{i} \right| \cdot \left| \boldsymbol{x}_{k,i}^{a\star} - \boldsymbol{b}_{i} \right|}$$
(3)

where  $x_k^{a\star}$  is the analysis mean state predicted by the LSTM, n is the number of state vector components  $x_{k,i}^{a\star}$  and  $a_i$  and  $b_i$  are the minimum and maximum physical bounds, respectively, of the i-th element of the state vector, defined as the minimum historical and maximum historical records.

Furthermore, any LSTM prediction that fell below zero was forced back to zero, effectively managing the intermittent nature of snow data.

This combined loss function is inspired by Physics-Informed Deep Learning (Cheng and Zhang, 2021), where domain-specific physical constraints guide the learning process.

## 275 2.4.3 Algorithm development and test configurations

The LSTM algorithm was trained using the analysis state vectors generated by the EnKF  $\boldsymbol{\cdot} \boldsymbol{x}_k^a$ - to predict the corrected analysis mean state vector,  $\boldsymbol{x}_k^{a\star}$ . As a supervised learning task, the training process utilized both input features and target outputs. The input features included meteorological forcing variables, the model's forecast mean state vector  $\boldsymbol{x}_k^f$ , and the observation vector, while the target outputs consisted of the analysis mean state vectors  $\boldsymbol{x}_k^a$  from the EnKF. To evaluate its effectiveness, the LSTM predictions were compared to the analysis state vectors generated by the EnKF. To develop the LSTM algorithm, we used Python 3.9.21 programming language and the open source libraries Keras v.2.10.0 (Chollet et al., 2015) and Scikit-learn v.1.1.1 (Pedregosa et al., 2011).

To assess the LSTM robustness and transferability, four experimental setups were tested:

## 1 Site-Specific LSTMs for State Correction







Seven LSTMs were independently trained and tested on each site to optimize hyperparameters. For the site with >95% missing SWE data (WFJ), the LSTM was trained using an observation vector containing only snow depth, which instead were not missing. Site-specific limits, derived from historical data, were applied to constrain the training process. Since the training process relies on a cost function that combines the RMSE with a penalty term enforcing physical bounds, the site-specific limits for each state component — namely, the dry and wet components of SWE, snow density, and albedo — were derived from historical data records. These records were pre-processed following the distribution adjustment and scaling procedures described in Section 2.4.1. Since direct historical observations of wet SWE were not available, we assumed this variable to be proportional to the ratio between LWC and total SWE, with dry SWE estimated as the complementary term.

The available data were split by continuous time spans, using the hydrological year (from the 1st of October to the 30th of September) as the reference unit. Specifically, first 80% of the data, in terms of hydrological years, was allocated for training and testing using a 4:1 ratio, while the remaining 20% was reserved for operational testing. In the operational setup, the framework combined S3M model prediction and state updating with the LSTM (see figure 2). At each time step  $t_k$ , the prior state vector  $\boldsymbol{x}_k^f = \mathrm{S3M}\left(\boldsymbol{x}_{k-1}^{a\star}\right)$  from the S3M model's forward simulation was provided as input to the LSTM, along with meteorological forcing and the observation vector  $\boldsymbol{y}_k$ . The LSTM outputs the updated analysis state vector  $\boldsymbol{x}_k^{\star}$ , which served as the initial condition for the subsequent S3M prediction step  $\boldsymbol{x}_{k+1}^f = \mathrm{S3M}\left(\boldsymbol{x}_k^{a\star}\right)$  and so on, cycling between S3M prediction and LSTM updating. The framework was validated using root mean square error (RMSE) metrics for snow depth and SWE between ground observations and model predictions. It is important to stress that, while the training phase was performed in the conventional way of training neural networks -meaning multiple timestep as input to obtain a sequence of outputs - the operational testing phase was performed giving to the LSTM trained models only one timestep at a time, to be coupled with the forward step of the cryospheric model. The metrics were computed for both the test and the operational set; while the first was used to set hyperparameters, the second was used to analyse the performance of the model.

## 2 Incorporating Memory to the Site-Specific LSTMs

The second test configuration introduced an additional feature component to call back on the use of the "long" memory component of the LSTM during the operational test phase. The memory component includes the forecast from the previous timestep  $\boldsymbol{x}_{k-1}^f$  as well as the meteorological forcing from the previous time step k-1 (relative to the current step k). The input vector I at time step k, is constructed as follows:

$$\mathbf{I}_{k} = \left[\mathbf{m}_{k}, \mathbf{m}_{k-1}, \mathbf{x}_{k-1}^{f}\right] \tag{4}$$

where:

- $\mathbf{m}_k \in \mathbb{R}^d$ : the vector of meteorological forcing variables at time step k where d=6 is the number of forcing variables.
  - $\mathbf{m}_{k-1} \in \mathbb{R}^d$ : the meteorological forcing at the previous time step k-1 (see fig (2) memory component element)
  - $\mathbf{x}_{k-1}^f \in \mathbb{R}^n$ : the model forecast at the previous time step k-1 (see fig (2) memory component element)

## 3 Testing Transferability of Site-Specific LSTMs

While in the Configuration 1, separate LSTM models were trained and tested individually on each site using only site-specific data, in Configuration 3, we assessed the spatial transferability of these site-specific models by applying each LSTM trained on the low data sparsity sites (NGK, KHT, RME, FMI-ARC) to new data from (i) the remaining 20% holdout portion of the low-sparsity sites not used during training, and (ii) high data sparsity sites (CDP and TRG). The WFJ site was excluded from this evaluation due to extensive gaps in its SWE time series. In this test we chose to use the LSTM setup with the best performances among prior tests, hence the one with memory components ( see point 2)

## 4 Multisite LSTM with Global Limits

A multisite LSTM was trained using data from the four low data sparsity sites (NGK, KHT, RME, FMI-ARC), with global scaling derived from the combined datasets. The training dataset comprised 80% of the data from these four sites, while the remaining 20% alongside all data from the high data sparsity datasets (CDP, WFJ, TRG) were used to test the model generalization capacity over water year type, using the operational setup. Data split was made by randomly sampling whole hydrological years. The water year types were classified based on the total snow depth and include wet years, dry years, and average conditions. Additionally, the results where also analysed comparing the performances per each site with the performances of the best site-specific LSTM-DA algorithm.

Site-specific EnKF results were always used as input for training the LSTM, even in the case of multisite LSTM testing the EnKFs used to generate the training data were always site-specific.

#### 2.4.4 LSTM structure and hyperparameters

In this study, we manually tuned the hyperparameters of the model, selecting the optimal configuration for each LSTM network. Below are the hyperparameters we fine-tuned:

#### - Batch size:

The batch size determines the number of training samples processed in a single forward and backward pass. A critical consideration when choosing the batch size is balancing computational efficiency with the quality of model outputs. To match the size of the observation datasets for each site, we used a standard batch size of 128 for the sites of KHT and NG, and we reduced it each time selecting the most suitable value for optimal training performance on all the other datasets (Bishop and Bishop, 2023).

## 345 **– Epochs:**




The number of epochs refers to the total number of complete passes through the training dataset. While a higher number

**Figure 2.** Operational Setup for deep data assimilation. This diagram illustrates the operational workflow for integrating observational data with the S3M (CIMA's Cryospheric Model) framework, through data assimilation via a Long-Short-Term-Memory neural network.

of epochs allows the model to better capture complex patterns in the data, it also increases the risk of overfitting and computational cost. After experimenting with various configurations, we set the number of epochs to 500, allowing for sufficient learning while balancing efficiency.

## 350 - Early Stopping:


Early stopping is a technique used to prevent overfitting by halting training when the validation performance fails to improve for a specified number of epochs. In our case, we set the patience to 100, meaning that training would terminate if no improvement was observed in the validation performance for 100 consecutive epochs (Prechelt, 2002).

## Initial Learning Rate:

The learning rate controls the step size during the optimization process. A higher learning rate accelerates convergence but may lead to instability, while a lower learning rate can slow down the learning process. Given the relatively small size of our datasets, we chose an initial learning rate of 0.01 to ensure rapid convergence during the early stages of training (Smith, 2015).

#### - Learning Rate Decay:

To enhance convergence stability and prevent overshooting, we applied a learning rate decay factor of 1.5 periodically throughout training. This decay reduces the learning rate over time, allowing the model to fine-tune its parameters more effectively in the later stages of training.

#### Dense Lavers:





Each LSTM network used a single dense layer as the output layer. This dense layer was used to map the LSTM outputs to a fixed-size state vector. The number of neurons in this layer was set to 4, corresponding to the required output dimensions for each network (Murphy, 2023).

## - Hidden LSTM Layers:

We employed two distinct LSTM architectures based on the data sparsity at different sites. For data-dense sites, we used a single LSTM layer followed by a dense output layer, resulting in a simple 2-layer architecture. This configuration was chosen under the assumption that the data contained enough patterns for the model to learn effectively without requiring excessive model depth. In contrast, for sparse sites, a deeper 3-layer LSTM architecture was implemented, which included two LSTM layers and a dense output layer. This approach aimed to capture more complex dependencies within the data, thereby improving the model's ability to learn from sparser temporal patterns (LeCun et al., 2015).

#### - Hidden units per LSTM Layer:

The number of hidden units in each LSTM layer determines the memory capacity of the model. For dense sites, the number was set to 500, allowing the model to learn from more intricate temporal dependencies. For sparse sites, the number was reduced to 100 to prevent overfitting, given the smaller and sparser datasets (Murphy, 2023).

A note to the reader: In the following section, the term LSTM refers to the computation of the analysis mean model state vector, denoted as  $\boldsymbol{x}_k^{a\star} \in \mathbb{R}^n$ , using the LSTM approach. On the other hand, the term EnKF refers to the computation of the analysis mean model state vector, denoted denoted as  $\boldsymbol{x}_k^a \in \mathbb{R}^n$ , using ensemble-based data assimilation via the EnKF scheme.

#### 3 Results

This section presents the results from the four configuration tests, based on the operational testing setup (see Fig. 2). Our objective was to replicate the actual algorithm coupling mechanism required in a real-time setup, where the LSTM is used at each time step k to perform filtering.

#### 385 3.1 Performance with varying data sparsity

At sites where data is plentiful (that is, available data cover more than 80% of the period of record: NGK, KTH, FMI-ARC, RME), the LSTM demonstrated robust performances, meaning that they were generally comparable to the original EnKF (Figure 3). This, however, came with a considerable nearly 70% decrease in computing time. For instance, one year of simulation

using the parallelized EnKF took on average 20 minutes, while using the trained LSTM took only 6 minutes. Only in the case of NGK site, the LSTM-DA was able to outperform both the open loop simulation and the EnKF-DA; At all the other dense sites (KTH, RME, FMI-ARC), the mean RMSE increase relative to the EnKF for SWE estimation made by site specific LSTMs was within 10 mm (Figure 3, panel e). Similarly, the mean RMSE increase- averaged across sites- compared to the EnKF for snow depth estimation made by site specific LSTMs was equal to 6 cm (Figure 3, panel f). The only exception is the site of FMI-ARC were the LSTM-DA still underperformed compared to the EnKF, although the absolute values of RMSE are 1 order of magnitude lower than the ones on the other sites. The bias analysis (Figure 3, panel g and h) showed that snow depth exhibited a near zero bias, while the LSTM tended to overestimate SWE compared to the EnKF. However, both patterns were consistent in the EnKF and in the S3M open loop.






In the case of datasets with high data sparsity (CDP, WFJ, TRG), the performance of the LSTM was markedly worse than the EnKF estimation of both SWE and snow depth (+50 mm RMSE for SWE and +19 cm RMSE for snow depth, Figure 4 panel e and f). On the other hand, the timing of SWE and snow depth peaks, as well as the magnitude of snow depth peaks, are generally captured correctly, even in these challenging data sparse scenarios (see fig.4 panels a,b,c,d). However, minor discrepancies were noted, even in the case of low data sparsity, including an underestimation of peak snow depth (Figures 3, panels c and d) and a slight temporal shift in the SWE peak (Figure 3, panel a).

Both the EnKF and LSTM networks improved SWE and snow depth predictions over the Open Loop model, at least in the case of low data sparsity; indeed the LSTM resulted in a reduction of 25 mm in RMSE for SWE, while the EnKF achieved a better reduction of 31 mm. On the other hand, in case of high data sparsity, the LSTM increased the RMSE by 15 mm, while the EnKF reduced the RMSE by 38 mm. For snow depth, the LSTM reduced RMSE by 4 cm in low sparsity, while the EnKF showed a greater reduction of 9 cm. Under high sparsity, the LSTM reduced RMSE by 8 cm, with the EnKF providing a larger reduction of 27 cm.

When it comes to evaluating the Kling-Gupta Efficiency (KGE) (Gupta et al., 2009), for sites with denser measurements (on average 0.72 for both SWE and snow depth), the values are comparable to those obtained with the EnKF-DA (on average 0.75 for SWE and 0.85 for snow depth), supporting the observed improvement trend over the open loop simulation(on average 0.75 for SWE and 0.68 for snow depth). Conversely, in the case of sparse datasets, the lower KGE values( on average -0.4 for SWE and 0.25 for snow depth) highlight the limitations of the LSTM in achieving performances comparable to the EnKF-DA (on average -0.5 for SWE and 0.35 for snow depth). Nevertheless, the LSTM still outperformed the open loop, which recorded even lower KGE scores of -0.50 for SWE and -0.06 for snow depth.

Overall, the LSTM demonstrates a reduction in bias compared to the Open Loop under low data sparsity conditions, with a bias reduction of 7 mm in SWE and 3 cm in snow depth (Figure 3, panel h). This improvement becomes even more pronounced in high data sparsity scenarios, where the bias decreases by 15.96 mm in SWE and 5 cm in snow depth (Figure 4, panel h). However, despite these improvements, the LSTM still exhibits a higher bias compared to the EnKF.

## 3.2 The role of the memory component





For datasets characterized by low data sparsity (NGK, KTH, FMI-ARC, RME), incorporating a memory component into the LSTM improved its ability to capture the seasonal dynamics of SWE and snow depth, particularly in accurately representing the timing and magnitude of peak SWE (see Figure 5, panels a and b). However, in some instances (see Figure 5, panels c and d), the memory component did not lead to a significant performance gain. Instead, it primarily acted as a smoother, dampening short-term fluctuations without substantially enhancing predictive accuracy. Additionally, no significant changes were observed in the snow depth estimation, with a mean RMSE increase of 6 cm compared to the EnKF (Figure 5, panel f).

When considering sites with high data sparsity (CDP, WFJ, TRG), a LSTM with the addition of a memory component improved both quantitative and timing estimations of peak SWE and peak snow depth, compared to the LSTM estimates without memory. In fact, we found a mean reduction in RMSE equal to 10 mm for SWE estimates and equal to 0.5 cm for snow depth estimates. However, for datasets with extremely high levels of missing data (e.g., 95%, WFJ and TRG – where the assimilated observations consist of manually measured SWE data, as detailed in the corresponding site references), improvements were still insufficient to obtain scores comparable to the EnKF (see Figure 6, panels e and f). Nevertheless, the introduction of the memory component reduced model instability and improved snowmelt timing, particularly at sites with sparse observations.

Overall, considering both scenarios, biases (Figure 6, panel g ) were not affected by the introduction of a memory component. The inclusion of a memory component narrowed the performance gap between the EnKF and LSTM compared to the Open Loop. For low sparsity, the LSTM reduced RMSE for SWE by 29 mm, while in high sparsity, it limited the increase in SWE RMSE to just 3 mm. In terms of snow depth, the LSTM reduced RMSE by 13 cm in low sparsity and by 7 cm in high sparsity. However, the EnKF still outperformed this LSTM configuration in both cases, highlighting its superior performance despite the added memory and runtime cost.

The KGE values, for both dense and sparse datasets, confirm that the memory component primarily acts as a smoother and enhances performance in most scenarios.

## 3.3 Spatial transferability

The LSTM trained on KHT emerged as the only one transferable across sites (Figure 7). For SWE estimation this LSTM showed small drops in performances across other sites below 20% and, in some cases, even a performances boost (see LSTM on FMI-ARC, RMSE AND TRG on Tab. 3). On the other hand, performance drops for snow depth estimation varied considerably, from 60% to -1 % (Tab. 3). Other LSTMs, such as those trained in NGK and FMI-ARC, performed less consistently, showing notable increases in RMSE when transferred to several sites. While recent studies (Kratzert et al., 2024) have strongly advocated for multi-basin training to achieve robust and generalizable LSTM streamflow models, we intentionally present the single-point case here for snow hydrology to establish a performance lower bound for snow spatial transferability—highlighting whether even such a constrained model can outperform the open loop and compare with traditional data assimilation approaches.

|         | SWE RMSE [mm]         |                   |                 |                        | Snow Depth RMSE [cm] |                            |                   |                   |                       |                 |
|---------|-----------------------|-------------------|-----------------|------------------------|----------------------|----------------------------|-------------------|-------------------|-----------------------|-----------------|
| Site    | RMSE <sub>LOCAL</sub> | $\Delta_{\%}$ KTH | $\Delta_\%$ NGK | Δ <sub>%</sub> FMI-ARC | $\Delta_\%$ RME      | RMSE <sub>LOCAL</sub> [cm] | $\Delta_{\%}$ KTH | $\Delta_{\%}$ NGK | $\Delta_{\%}$ FMI-ARC | $\Delta_\%$ RME |
| NGK     | 14.09                 | +8                | -               | +125                   | +199                 | 8                          | +34               | -                 | +90                   | +126            |
| KHT     | 14.10                 | -                 | +271            | +329                   | +155                 | 22                         | -                 | -13               | +34                   | +2              |
| FMI-ARC | 9.06                  | -45               | +119            | -                      | +32                  | 11                         | +47               | +25               | -                     | +78             |
| RME     | 39.92                 | -51               | +35             | +59                    | -                    | 17                         | -54               | +14               | +7                    | -               |
| CDP     | 67.61                 | +18               | +68             | +74                    | +66                  | 12                         | +60               | +62               | +253                  | +128            |
| TRG     | 73.70                 | -76               | -58             | -37                    | -68                  | 22                         | -1                | +2                | +19                   | +11             |
| Average | -                     | -29               | +87             | +110                   | +77                  | -                          | +17               | +18               | +81                   | +69             |

Table 3. Percentage change in snow water equivalent and snow depth RMSE when using a transferred LSTM assimilation scheme compared to a locally trained LSTM. Positive and negative values indicate improvements or degradation in performance, respectively.  $\Delta$  values are obtained as the difference between the RMSE of a locally trained LSTM and that of a transferred LSTM, respectively for SWE and snow depth.

Tests on correlations between LSTMs performances and biases with various climatological variables showed no statistically significant correlation (see fig. A1 and A2 in the appendix).

## 455 3.4 Multi-site Long-Short Term Memory

To guarantee a meaningful and practical evaluation of the multi-site LSTM performances, the analysis was performed by comparing RMSE distributions for SWE and snow depth across water year types. Figure 8 presents the RMSE distribution for SWE and snow depth under varying water year types, comparing the performance of the S3M open-loop run, the estimates retrieved from the analysis of EnKF, and the LSTM estimates.

A multi-site LSTM generally demonstrated improvements in performance compared to the S3M open-loop run, particularly for SWE. For dry and average years (Figure 8, panels b and c), the SWE estimates from the LSTM showed competitive performance over EnKF, with a performance drop of less than 6 mm on average. On the other hand, the LSTM SWE estimation RMSE values were higher during wet years (+15 mm). Reduced performances of the Multi-site LSTM simulation on SWE over wet years may be because in wet years, an increased number of snowfall events may introduce additional complexity and uncertainty, both due to the cascading effects of uncertainties in initial conditions and precipitation phase partitioning (Harder and Pomeroy, 2014). Moreover, the formation of several snow layers may not be fully captured by S3M.

For snow depth, the improvements were less clear across all water year types. The RMSE reduction remained modest, with an average loss of 1.8 cm.

Comparing the multi-site LSTM DA with the site-specific LSTM DA trained over KHT, results show comparable performance for SWE, with neither approach consistently outperforming the other (see Fig. 9). In some cases, the site-specific model achieves lower errors, while in others the multi-site model performs equally well or slightly better. For snow depth, however, the multi-site LSTM DA tends to outperform the site-specific LSTM DA across most sites, although the improvements are generally modest (e.g. see Fig. 9 pannel d).

## 4 Discussion





In snow-dominated regions, accurate snow estimations are crucial for water resources managing, floods forecasting (Andreadis and Lettenmaier, 2006), and for assessing the impact of climate change on the hydrological cycle (Siirila-Woodburn et al., 2021). Nonetheless, significant uncertainties in model predictions and observational data make accurate snow estimates challenging (Blöschl, 1999). Data assimilation, which integrates both sources, is arguably one of the most effective methods for improving snowpack-model reliability. However, state-of-the-art ensemble-based techniques like the EnKF are computationally intensive, potentially limiting their use in operational contexts. Furthermore, one can argue that it is not just the computational expense but also the time and effort required for parameter tuning, setup, and execution that pose significant challenges to their widespread adoption in such applications.

This paper suggests an alternative assimilation framework for snow, which relies on having a LSTM neural network (Adnan et al., 2024; Song et al., 2024) to learn how to perform the filtering updates performed by an EnKF.

The key hypothesis underlying this research was that, leveraging Deep Learning, it is possible to preserve the skill of an EnKF, while significantly reducing computational efforts. This paper outlined four key findings in this regard.

First, site-specific LSTMs achieved comparable performances to an EnKF, both in predicting SWE and snow depth, as well as their seasonal patterns, with also a significant reduction in computational time. Besides this temporal efficiency, the LSTM enabled leveraging a complex tool like the EnKF only for initial training, then replicating its capabilities in operational settings using a faster, simpler data assimilation framework.

To evaluate the computational efficiency of the proposed framework, it was benchmarked against a parellized EnKF. Even though the EnKF already benefits from parallelization during the ensemble prediction step using 15 CPU cores, once trained the LSTM-based approach provided a further 70% reduction in computational time. This result underscores the potential of the framework to significantly lower computational overhead, particularly in scenarios with limited resources or parallelization capabilities.

In line with the work of Guidicelli et al. (2024), this finding reinforces the potential of Deep Learning for data assimilation in snow hydrology. Yet, the LSTM performance was found to be highly sensitive to the temporal resolution of the input data, which is consistent with findings from other machine learning studies (Xu and Liang, 2021; Gong et al., 2023). These results emphasize the importance of acquiring high-frequency snow data to ensure optimal performance and accuracy of modern data-assimilation approaches (Dedieu et al., 2016), highlighting the need for investments in this direction (Cui et al., 2023).

Second, the introduction of memory into the algorithm improved both stability and performance, particularly when working with the inherently noisy outputs of the EnKF and in locations where data sparsity was a major issue. Future efforts could explore additional pre-processing of input data to reduce noise (e.g., smoothing or moving averages), though care must be taken to preserve snow intermittency, which is critical in certain hydrological contexts.

Third, the LSTM trained on a long dataset (KHT) demonstrated some potential for spatial transferability with minimal performance loss, opening avenues for distributed applications of deep data assimilation provided that such long datasets are used in training. Although using limited datasets in both temporal and spatial coverage compared to recent studies (Song

et al., 2024), our approach proved to be effective in speeding up traditional data assimilation techniques while maintaining comparable performance. Additionally, our framework, designed to test the operational viability of a quasi-real-time pilot point method, still proved the feasibility of an alternative use of LSTM algorithm without loss in performances. The encouraging results provide a foundation for extending this framework to broader, more diverse networks in future research. The lack of statistically significant correlations between performance and specific climatological variables further supports transferability. According to Karniadakis et al. (2021), Deep Learning, which usually requires a large amount of data to optimally generalize over samples, has a stronger generalization capability, even in small data regimes, if such algorithms are developed with a physics-informed learning approach. In light of this, we introduced soft physical constraints into the cost function as a way to incorporate an inductive bias. Although this particular approach did not prove effective in significantly enhancing generalization, considerable potential remains in enforcing snow physical constraints in LSTMs (Charbonneau et al., 2024). Further research is needed in this direction to better understand how such constraints can support model generalization and physical consistency. This finding could contribute to the ongoing debate around the unresolved question of Deep Learning models transferability (Pakdehi et al., 2024).

With few exceptions, the comparison of RMSE reductions from the Open Loop to the analysis of the LSTM demonstrated substantial improvements. All but 2 out of the 14 site-specific LSTM frameworks significantly outperformed the Open Loop, although none outperformed the EnKF. Nevertheless, the LSTM ability to deliver marked improvements over the Open Loop underscores its promise as a computationally efficient and effective alternative, even under challenging conditions.

Regularization in particular, and uncertainty-quantification more generally, could be improved by using Bayesian Deep Learning (Murphy, 2023). For example, a recent cryospheric study used an ensemble Kalman method (rather than stochastic gradient descent) to train Bayesian neural network with an architecture that was tailored to the problem at hand (Pirk et al., 2024). The contrast between our study, where a neural network learns to mimic the EnKF update, and Pirk et al. (2024) where an EnKF method trains an uncertainty-aware neural network, are just some recent examples of the synergies that exist between Bayesian data assimilation and Deep Learning. The aforementioned study of Guidicelli et al. (2024) also explored how DA and Deep Learning could be combined for better uncertainty quantification, not only by having a neural network learn the posterior spread from an EnKF method but also by adopting a simple dropout technique for approximate uncertainty quantification in the neural network outputs. The links between Deep Learning and Bayesian data assimilation are well established in the literature (Arcucci et al., 2021; Cheng et al., 2023; Murphy, 2023), but we emphasize them once more in this discussion because they are perhaps less known to the snow science community.

The fourth and last key aspect that this study highlighted was no dependency of the performance of this algorithm on dry and average water years, despite a diminished robustness in wet years. Nonetheless, this limitation is shared with both the EnKF and S3M in open-loop, as shown by the distributions in these scenarios. Given the predicted decline in snow cover over the coming decades and the emergence of more frequent snow droughts (Larsson Ivanov et al., 2022), the reduced performance of the algorithm in wet years may have a relatively minor overall impact. Under such a fast-paced changing climate, a climatically robust LSTM could account for physical processes changing faster than scientists change their models (Cowherd et al., 2024a). Additionally, considering a comparison between two approach, neither the site-specific nor the multi-site LSTM-DA

consistently outperforms the other. While multi-site training is theoretically expected to improve generalization by exposing the model to a broader range of conditions, this benefit is not clearly observed for SWE. A likely reason could be an uneven representation of sites, combined with variability in snowpack properties, meteorological drivers, and measurement methods, which may bias the model and introduce noise, leading to underfitting. Snow density also plays a crucial role; SWE is defined as  $W = d\rho$ , where W is SWE [kg/m²], d is snow depth [m], and  $\rho$  is bulk snow density [kg/m³]. A site-specific model such as KHT may implicitly capture a representative density evolution that transfers well across sites, whereas a multi-site model must attempt to generalize density dynamics across all environments, often with less accuracy. Overall, the multi-site LSTM-DA and the EnKF-DA perform similarly, with the latter only marginally better. This is encouraging, as it highlights the potential of the multi-site LSTM-DA to achieve comparable performance while substantially reducing the computational cost associated with ensemble-based methods.

It is important to note that the algorithm showed a significant drop in performance when handling missing or sparse data, contrary to an Ensemble Kalman Filter. Future work in this regard should focus on improving performance under circumstances of high data sparsity, exploring advanced smoothing techniques, and extending transferability even to ungauged catchments. Finally, while our results are based on high-quality forcing and observational datasets, we acknowledge that operational applications may involve lower-quality inputs. In such cases, pre-processing strategies (e.g., bias correction, gap-filling) and hybrid DA–AI frameworks could help mitigate performance loss, with the potential to selectively down-weight unreliable inputs rather than propagating their errors through the model. Recent work by Gauch et al. (2025) demonstrates the effectiveness of imputation and correction methods for handling missing or degraded data in operational environments, while generative models such as those explored by Dhoni (2023) offer promising avenues for enriching and augmenting incomplete datasets.

#### 5 Conclusions







We proposed a data assimilation framework based on Deep Learning, leveraging a LSTM to perform data assimilation for state estimation in a hybrid temperature-and-radiation driven hydrology-oriented cryosphere model. The LSTM framework showed performances in snow depth and SWE estimation that were comparable to an EnKF, while significantly reducing computational time. Furthermore, a LSTM trained on a long dataset, proved to be spatially transferable, with only a ~20% reduction in SWE estimation performance when applied to regions outside the training domain. LSTM robustness during dry and average water years further underscores the generalization capacity of such a framework. Using the LSTM as an emulator of the ensemble Kalman Filter allows us to significantly reduce the computational cost of ensemble-based data assimilation. This is particularly advantageous in a spatially distributed configuration, where running a full ensemble over large domains could otherwise be prohibitively expensive. In our current setup, ensembles are required only during the LSTM training phase, not at inference time — resulting in a more efficient approach for operational use.

Preliminary tests with multi-site LSTM configurations have shown promising results: a single LSTM model trained on data from multiple locations can generalize well to other sites. Building on this idea, we envision extending the application of

LSTMs from single-point setups to multiple representative points within a catchment. This spatially sparse assimilation could then be combined with an interpolation or spatial mapping strategy to propagate the correction across the entire domain.

Such an approach would provide a practical compromise between the need for spatially distributed corrections and the computational limitations of full-domain deep data assimilation. This transition from point-based to distributed correction—leveraging spatial generalization and interpolation—will be a key focus of our future work. The LSTM, however, showed limitations when dealing with sparse data scenarios. Addressing these limitations could involve exploring advanced smoothing techniques to be applied to input data or evaluate the benefit from merging different kinds of data sources (e.g., remotely sensed data). These results open a window of opportunity for spatially distributed deep data assimilation; hence future work should focus on testing such a spatio-temporal configuration. Moreover, it would be valuable to assess the impact of Deep Learning in the assimilation of snow data for water resources applications, such as streamflow estimation. This study contributes to the relatively under-explored literature on Deep-Learning-based data assimilation by suggesting Deep Learning algorithms as efficient and computationally less intensive data assimilation frameworks for operational snow hydrology.

Code availability. The S3M snow model is available at the CIMA Foundation's Hydrology and Hydraulics repository at https://github.com/c-hydro/s3m-dev (last access: 21 January 2025). S3M is also available on Zenodo at https://doi.org/10.5281/zenodo.4663899 (Avanzi and Delogu,2021).

Data availability. Sources of data used in this paper are reported in Section 2.1. Data from the site of Nagaoka were provided by the Snow and Ice Research Center, National Research Institute for Earth Science and Disaster Resilience, Nagaoka, Japan.

Author contributions. GB, FA, and LC conceived the investigation, with contributions from all coauthors. GB worked on the development and implementation of the algorithm and carried out the analyses for the validation. All authors were involved in discussing the results, reviewing the manuscript, and drafting the final version.

Competing interests. One author is a member of the editorial board of The Cryosphere

Acknowledgements. Part of this research was performed under the Memorandum of Cooperation between CIMA Research Foundation (Italy) and the Snow and Ice Research Center, National Institute for Earth Science and Disaster Resilience (Japan). KA acknowledges funding from the ERC-2022-ADG under grant agreement No 01096057 GLACMASS and an ESA CCI Research Fellowship (PATCHES project). MG acknowledges funding from the NASA Understanding Changes High Mountain Asia Program (GRANT 80NSSC20K1301). The Torgnon (Italy) and Col de Porte (France) sites are part of the International Network for Alpine Research Catchment Hydrology (INARCH).

**Figure 3.** Results for sites with low data sparsity, site spefic LSTM. Panels a,b,c,d: comparison between ground observation (red) of Snow Water Equivalent (top) and snow depth (bottom) and model estimates by S3M in the open loop (black), using an Ensemble Kalman filter (grey), and using a Long Short Term Memory neural network (blue) in Kuhtai (row 1) and Nagaoka (row 2). Panels e,f,g,h,i,j: box plots of RMSE, bias and KGE for SWE (pan.e for RMSE, panel f for bias and panel i for KGE) and snow depth (pan.g for RMSE, panel h for Bias and panel j for KGE); points represent sites with less than 3 years of validation data.

**Figure 4.** Results for sites with high data sparsity, site spefic LSTM. Panel a,b,c,d: comparison between ground observation (red) of Snow Water Equivalent (top) and snow depth (bottom) and model estimates by S3M in open loop (black), using an Ensemble Kalman filter (grey), and using a Long Short Term Memory neural network (blue) in Col de Porte (row 1) and Weissfluhjoch (row 2).Panels e,f,g,h,i,j: box plots of RMSE, bias and KGE for SWE (panel e for RMSE, panel f for bias and panel i for KGE) and snow depth (panel g for RMSE, panel h for Bias and panel j for KGE); points represent sites with less than 3 years of validation data.

**Figure 5.** Results for sites with low data sparsity. Panel a,b,c,d: comparison between ground observation (red) of Snow Water Equivalent (top) and snow depth (bottom) and model estimates by S3M in open loop (black), using an Ensemble Kalman filter (grey), using a Long Short Term Memory neural network with memory (light blue) in Kuhtai(row 1) and Nagaoka (row 2). Panels e,f,g,h,i,j: box plots of RMSE, bias and KGE for SWE (panel e for RMSE, panel f for bias and panel i for KGE) and snow depth (panel g for RMSE, panel h for Bias and panel j for KGE); points represent sites with less than 3 years of validation data.

**Figure 6.** Results for sites with high data sparsity. Panel a,b,c,d: comparison between ground observation (red) of Snow Water Equivalent (top) and snow depth (bottom) and model estimates by S3M in open loop (black), using an Ensemble Kalman filter (grey), using a Long Short Term Memory neural network with memory (light blue) in Col de Porte (row 1) and Weissfluhjoch (row 2). Panels e,f,g,h,i,j: box plots of RMSE, bias and KGE for SWE (pan.e for RMSE, panel f for bias and panel i for KGE) and snow depth (pan.g for RMSE, panel h for Bias and panel j for KGE); points represent sites with less than 3 years of validation data.

**Figure 7.** Spatial transferability of site-specific LSTMs for SWE and snow depth estimation. Panel (a) shows a comparison between the RMSE for SWE obtained by using each LSTM at the training site (x-axis) and the RMSE obtained when transferring the same LSTM to other sites (y-axis). Panel (b) shows the same information, but for snow depth. The bisectors in the two panels represent the one-to-one lines comparing the RMSE values for SWE and between the site-specific LSTM and the LSTM trained on a different site. The dotted lines in both panels serve as benchmarks, indicating the RMSE values achieved by the site-specific LSTM models. Colors represent training sites, while shapes correspond to the to sites where each LSTM was applied. The lowest granularity site, WFJ, is excluded.

**Figure 8.** RMSE distribution for SWE and snow depth across water year types RMSE distribution for SWE on wet, dry and average years type (panels a,b,c) and snow depth (panels d,e,f) under varying water year types: wet, dry, and average conditions.

## Comparison of RMSE between Multi-site and Site-specific LSTM-DA across site

**Figure 9.** Comparison of RMSE for snow water equivalent (SWE, left column) and snow depth (right column) across multiple sites and methods. Results are shown for LSTM-DA multisite (blue), S3M open loop (black), EnKF-DA (grey), and LSTM-DA site-specific (orange).

## **Appendix A: Coordinates information of the 7 study sites**

- TRG (Torgnon, Aosta Valley, Italy): 45°50′ N, 7°34′ E
- CDP (Col De Porte, Isère, France): 45°3′ N, 5°77′ E
- WFJ (Weissfluhjoch, Davos, Switzerland): 46°82′ N, 9°8′ E
- KHT (Kühtai, Tirol, Austria): 47°20′71″N, 11°00′6″E
  - FMI-ARC (FMI-ARC Sodankylä Geophysical Observatory, Finnish Lapland): 67°36′8″ N, 26°63′3″ E
  - NGK (Nagaoka, Japan): 37°25′ N, 138°53′ E
  - RME (Reynolds Mountain East, Idaho, USA): 43°11′9.36″ N, 116°46′58.9″ W

| Site    | SWE Obs. (mm)                      | HS Obs. (cm) | Frequency | Error (SWE/HS) [mm/cm] | Time Range        |
|---------|------------------------------------|--------------|-----------|------------------------|-------------------|
| TRG     | 6h, missing (2012–2013, 2014–2015) | ✓            | 30'       | ±15/±10                | Oct 2012–Mar 2023 |
| CDP     | From 2002                          | ✓            | 1h        | ±5/±1                  | Oct 1993–Sep 2022 |
| WFJ     | Manual, sporadic                   | ✓            | 60'       | ±10/±20                | Oct 1999–Sep 2018 |
| KHT     | ✓                                  | ✓            | 15'       | ±1/±10                 | Oct 1990–Sep 2015 |
| FMI-ARC | Manual, sporadic                   | ✓            | 60'       | ±15/±10                | Oct 2007–Jul 2014 |
| NGK     | ✓                                  | ✓            | 60'       | ±10/±10                | Oct 2006–Aug 2023 |
| RME     | ✓                                  | From 1999    | 60'       | ±10/±10                | Oct 1984–Sep 2008 |

**Table A1.** Measurement Characteristics Across Sites. TRG = Torgnon, Aosta Valley, Italy. CDP = Col de Porte, Isère, France. WFJ = Weissfluhjoch, Davos, Switzerland. KHT = Kühtai, Tirol, Austria. FMI-ARC = FMI-ARC Sodankylä Geophysical Observatory, Finnish Lapland. NGK = Nagaoka, Japanel RME = Reynolds Mountain East, Idaho, USA.

#### SITE CHARACTERISTICS INDEPENDENCY **RMSE SWE vs Peak SWE RMSE Snow Depth vs Peak SWE** Corr: 0.75 P-value: 0.05 Corr: 0.76 P-value: 0.05 Corr: 0.45 P-value: 0.32 Corr: 0.45 P-value: 0.31 (a) (b) **LWSE [cm]** 0.2 RMSE [mm] 00 0.2 50 300 700 800 200 300 500 600 700 800 200 400 500 600 400 Peak SWE [mm] Peak SWE [mm] RMSE SWE vs Altitude **RMSE Snow Depth vs Altitude** Corr: 0.82 P-value: 0.02 Corr: 0.90 P-value: 0.01 Corr: 0.75 P-value: 0.05 (c) (d) **EMSE [cm]** 0.2 RMSE [mm] Corr: 0.73 P-value: 0.06 00 0.2 50 0 2000 500 1500 2000 2500 500 1000 1500 2500 1000 Ò Altitude [m] Altitude [m] **RMSE SWE vs MAP RMSE Snow Depth vs MAP** Corr: 0.05 P-value: 0.91 Corr: 0.00 Corr: -0.32 P-value: 0.48 Corr: -0.58 (e) (f) **RMSE** [cm] 0.2 RMSE [mm] P-value: 0.99 P-value: 0.18 0.2 50 1500 1500 500 1000 2000 2500 500 1000 2000 2500 Mean annual precipitation [mm] Mean annual precipitation [mm] **RMSE SWE vs Latitude** RMSE Snow Depth vs Latitude Corr: -0.25 P-value: 0.59 (g) Corr: -0.01 P-value: 0.98 (g) **RMSE** [cm] 0.2 RMSE [mm] 100 Corr: -0.22 Corr: -0.01 50 65 40 45 50 55 60 45 50 55 65 40 60 Latitude [°] Latitude [°] **RMSE SWE vs Longitude RMSE Snow Depth vs Longitude** Corr: -0.20 P-value: 0.67 Corr: -0.25 P-value: 0.59 Corr: -0.22 P-value: 0.63 Corr: -0.58 P-value: 0.17 (h) **(I) EW 0.2** 0.2 0.1 RMSE [mm] 00 50 -50 Ò 50 100 50 100 150 -100150 -100-50Ó Longitude [°] Longitude [°] LSTM-DA No Memory LSTM-DA with Memory

Figure A1. Analysis of RMSE dependency on site characteristics for SWE and snow depth across different parameters. Subplots (a-b) show RMSE vs. peak SWE, (c-d) vs. altitude, (e-f) vs. annual precipitation, (g-h) vs. latitude, and (i-l) vs. longitude. Blue and cyan markers represent estimations from LSTM with and without memory, respectively. Correlation coefficients, confidence intervals, and p-values indicate weak or negligible dependence of RMSE on these site characteristics, suggesting general independence of model performance from these factors.

#### SITE CHARACTERISTICS INDEPENDENCY **Bias SWE vs Peak SWE** Bias Snow Depth vs Peak SWE Corr: 0.34 P-value: 0.46 Corr: -0.06 Corr: -0.64 P-value: 0.12 Corr: -0.68 P-value: 0.10 (a) (b) Bias [mm] 0.05 Bias [cm] P-value: 0.89 0 0.00 -0.05-50 200 300 700 800 200 300 500 700 800 400 500 600 400 600 Peak SWE [mm] Peak SWE [mm] Bias SWE vs Altitude **Bias Snow Depth vs Altitude** Corr: -0.16 P-value: 0.73 Corr: -0.15 P-value: 0.75 (c) (d) Corr: 0.26 P-value: 0.58 Bias [mm] 0.05 Bias [cm] 0 0.00 -0.05 -50Ö 500 1500 2500 500 2000 2500 1000 2000 Ó 1000 1500 Altitude [m] Altitude [m] **Bias SWE vs MAP** Bias Snow Depth vs MAP Corr: -0.41 P-value: 0.36 Corr: -0.42 Corr: 0.01 P-value: 0.98 Corr: 0.17 (e) (f) 0.05 Bias [mm] Bias [cm] P-value: 0.34 P-value: 0.72 0 0.00 -0.05-50 1000 1000 2000 500 1500 2000 500 2500 1500 2500 Mean annual precipitation [mm] Mean annual precipitation [mm] **Bias SWE vs Latitude** Bias Snow Depth vs Latitude Corr: -0.41 P-value: 0.36 (g) (h) Bias [mm] 0.05 Corr: -0.01 Corr: -0.35 Bias [cm] P-value: 0.98 P-value: 0.44 0 0.00 -0.05-50 45 55 65 40 50 60 65 40 45 50 55 60 Latitude [°] Latitude [°] Bias SWE vs Longitude **Bias Snow Depth vs Longitude** Corr: -0.01 P-value: 0.98 Corr: 0.37 P-value: 0.42 Corr: -0.11 P-value: 0.81 Corr: -0.15 P-value: 0.75 (i) **(I)** Bias [mm] 0.05 Bias [cm] 0 0.00 -0.05-50 -50 -100Ó 50 100 -100 -50Ó 50 100 150 150 Longitude [°] Longitude [°] LSTM-DA with Memory LSTM-DA No Memory

**Figure A2.** Bias analysis of SWE and snow depth with respect to site characteristics. Subplots (a-b) illustrate bias vs. peak SWE, (c-d) vs. altitude, (e-f) vs. annual precipitation, (g-h) vs. latitude, and (i-l) vs. longitude. Blue and cyan markers represent estimations from LSTM without and with memory, respectively. Correlation coefficients and p<sup>33</sup>alues suggest minimal or no significant bias dependency on these site characteristics, except for a moderate correlation in specific cases, such as SWE bias with annual precipitation in (e).

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
