# Peer review of "Learning to filter: Snow data assimilation using a Long Short-Term Memory network"

_EGUsphere, 2025_

## Author Comment (AC1)

**General comment**

*This work developed a surrogate for EnKF-DA using an LSTM network. The introduction and methods sections are well written and structured. However, there are several errors in the results that are inconsistent with the plots. More importantly, the results lack sufficient explanation and analysis regarding why the LSTM performs differently from EnKF at different sites or scenarios. The discussion could benefit from additional comparisons with previous studies and a deeper analysis of the results. Currently, it leans more toward reinforcing the need for LSTM in data assimilation, which somewhat repeats points already made in the introduction. Therefore, I recommend a major revision before publication.*

We thank the reviewer for their helpful comments. We appreciate the positive feedback on the introduction and methods, and we acknowledge the concerns raised. To improve our manuscript clarity and coherence we will modify part of the results sections and update Figures 3–6 to better align with the text, and we will add additional clarification around the structure of the LSTM algorithm. Please find in the following our response to each specific comment.

**Specific Comments**

**Line 103–105:** *What is the source of the meteorological forcing data? Are they derived from gridded datasets?*

The meteorological forcing data used in this study are point-based and specific to each site. Further details are available in the references cited in Section 2.1.

**Table 2:** *The data time span for each site should be mentioned.*

To maintain a concise structure, we have included this information in the Appendix (see Table A1).

**Line 171:** *Forecasted model state is $x_k^f$.*

We acknowledge the ambiguity and will revise the sentence at line 171 as follows for clarity:

> The Kalman gain $\mathbf{K}_k$ in Equation (2) acts as a weighting factor, balancing the correction term (the innovation) by accounting for the relative uncertainties in the forecasted model state through the forecast error covariance matrix $\mathbf{P}_k^f$ and in the observations through the observation covariance matrix $\mathbf{R}_k$.

**Line 277–278:** *Please clarify how the data were split: by individual data points or by continuous time spans?*

We split the data by continuous time spans using hydrological years (October 1st – September 30th). The revised sentence will read:

> The available data were split by continuous time spans, using the hydrological year (from the 1st of October to the 30th of September) as the reference unit.

Specifically, the first 80% of the data, in terms of hydrological years, was allocated for training and testing using a 4:1 ratio, while the remaining 20% was reserved for testing.

**Line 276:** *Please clarify what are site-specific limits here*

We agree with the reviewer that the information given could benefit from additional explanation. Hence, we plan to add this sentence at line 276 :

Since the training process relies on a cost function that combines the RMSE with a penalty term enforcing physical bounds, the site-specific limits for each state component — namely, the dry and wet components of SWE, snow density, and albedo - were determined as physical bounds derived from historical data records. The historical records were initially pre processed following the distribution adjustment and scaling procedures described in Section 2.4.1.

**Line 288–290:** *Please use a formula to clarify this configuration. Do you mean that $x_k^f$ and forcing at both time steps k and k-1 are used as LSTM inputs in the second test? Please refer to Figure 2 for clarity.*

We plan to add a clarifying formula at Line 290. Here it is the suggested clarification :

In this second test configuration, the input vector I at time step $k$, is constructed as follows:

$$\mathbf{I}_k = \left[\mathbf{m}_k \ \mathbf{m}_{k-1} \ \mathbf{x}_{k-1}^{f\star}\right] \tag{1}$$

where:

- $\mathbf{m}_k \in \mathbb{R}^d$: the vector of meteorological forcing variables at time step $k$ where $d = 6$ is the number of forcing variables.

- $\mathbf{m}_{k-1} \in \mathbb{R}^d$: the meteorological forcing at the previous time step $k-1$ (see fig (2) memory component element)

- $\mathbf{x}_{f_{k-1}^*} \in \mathbb{R}^n$: the model forecast at the previous time step $k-1$(see fig (2) memory component element)

**Line 292–294:** *This part is confusing. What is the difference between Configuration 1 and Configuration 3? Was a single LSTM selected from Configuration 1 and then applied to other sites? Please clarify.*

We thank the reviewer for this helpful comment towards improving the comprehension of our work. To improve the manuscript we plan to re-write section 2.4.3 point 3 as follows :

While in the Configuration 1, separate LSTM models were trained and tested individually on each site using only site-specific data, in Configuration 3, we assessed the spatial transferability of these site-specific models by applying each LSTM trained on the low data sparsity sites (NGK, KHT, RME, FMI-ARC) to new data

from (i) the remaining 20% holdout portion of the low-sparsity sites not used during training, and (ii) high data sparsity sites (CDP and TRG). The WFJ site was excluded from this evaluation due to extensive gaps in its SWE time series. In this test we chose to use the LSTM setup with the best performances among prior tests, hence the one with memory components ( see point 2)

**Line 299–300:** *Is there a specific reason to randomly sample water years for data splitting rather than using a continuous historical time span to train the model and a continuous future time span to test it? Random sampling can create artificially easier test conditions by allowing test data (time period) to fall between training water years, which may provide the model with indirect information about future conditions.*

We understand the reviewer remark. We chose to randomly sample water years to develop a statistically robust algorithm with improved transferability across both temporal and spatial domains. Our goal was to avoid overfitting to long-term climate trends that may be present in a continuous historical time span, but that may not be representative of a warming future.

At the same time, by training and testing on entire water years, we ensure that snowpack conditions reset annually, eliminating inter-annual dependencies. Finally, since we work with reanalysis data and aim to implement this as an operational tool, maintaining a strict future–past separation is less critical. Instead, our priority is to enhance model generalization and robustness across diverse conditions while minimizing bias from long-term temporal correlations in the training set.

***LSTM structure and hyperparameters were not mentioned in this work.***
We plan to add a paragraph at section 2.4:

In this study, we manually tuned the hyperparameters of the model, selecting the optimal configuration for each LSTM network. Below are the hyperparameters we fine-tuned:

- **Batch size:**
  The batch size determines the number of training samples processed in a single forward and backward pass. A critical consideration when choosing the batch size is balancing computational efficiency with the quality of model outputs. To accommodate the size of the observation datasets for each site, we used a standard batch size of 128 for the sites of KHT and NG, and we reduced it each time selecting the most suitable value for optimal training performance on all the other datasets (Bishop & Bishop 2023).

- **Epochs:**
  The number of epochs refers to the total number of complete passes through the training dataset. While a higher number of epochs allows the model to better capture complex patterns in the data, it also increases the risk of overfitting and computational cost. After experimenting with various configurations, we set the number of epochs to 500, allowing for sufficient learning while balancing efficiency .

- **Early Stopping Patience:**
  Early stopping is a technique used to prevent overfitting by halting training

when the validation performance fails to improve for a specified number of epochs. In our case, we set the patience to 100, meaning that training would terminate if no improvement was observed in the validation performance for 100 consecutive epochs(Prechelt 2002).

- **Initial Learning Rate:**
  The learning rate controls the step size during the optimization process. A higher learning rate accelerates convergence but may lead to instability, while a lower learning rate can slow down the learning process. Given the relatively small size of our datasets, we chose an initial learning rate of 0.01 to ensure rapid convergence during the early stages of training(Smith 2015).

- **Learning Rate Decay:**
  To enhance convergence stability and prevent overshooting, we applied a learning rate decay factor of 1.5 periodically throughout training. This decay reduces the learning rate over time, allowing the model to fine-tune its parameters more effectively in the later stages of training.

- **Dense Layers:**
  Each LSTM network used a single dense layer as the output layer. This dense layer was used to map the LSTM outputs to a fixed-size state vector. The number of neurons in this layer was set to 4, corresponding to the required output dimensions for each network(Murphy 2023).

- **Hidden LSTM Layers:**
  We employed two distinct LSTM architectures based on the data sparsity at different sites. For dense sites, a single LSTM layer was used, resulting in a simple 2-layer architecture. This configuration was chosen under the assumption that the data contained enough patterns for the model to learn effectively without requiring excessive model depth. In contrast, for sparse sites, a deeper 3-layer LSTM architecture was implemented, which included two additional LSTM layers. This approach aimed to capture more complex dependencies within the data, thereby improving the model's ability to learn from sparser temporal patterns(LeCun et al. 2015).

- **Hidden units per LSTM Layer:**
  The number of hidden units in each LSTM layer determines the memory capacity of the model. For dense sites, the number was set to 500, allowing the model to learn from more intricate temporal dependencies. For sparse sites, the number was reduced to 100 to prevent overfitting, given the smaller and sparser datasets(Murphy 2023).

**Line 309–311 (Figure 3):** *Is this result from testing or operational testing? Please clarify*

To clarify that the results refer to an operational testing, we plan to add this sentence at the beginning of the section 3:

> This section presents the results from the four configuration tests, based on the operational testing setup (see Fig. 2). Our objective was to replicate the actual

algorithm coupling mechanism required in a real-time setup, where the LSTM is used at each time step $k$ to perform filtering.

**Line 313–314:** *It is somewhat difficult to distinguish the EnKF-DA and LSTM boxes in the plots. If the last box in each panel represents LSTM-DA, it suggests that the RMSE values of LSTM-DA for KHT, RME, and FMI-ARC increased compared to EnKF-DA, with KHT showing the largest increase. This appears inconsistent with the narrative presented here. Please check.*

We apologize for the low quality of the figure and the inconsistency with the text; we plan to modify the figures from 3 to 6 and particularly the sentence at line 313-318 to adhere more to the graph. The new version will be :

> Only in the case of NGK site, the LSTM-DA was able to outperform both the open loop simulation and the EnKF-DA; At all the other dense sites ( KTH, RME, FMI-ARC), the mean RMSE increase relative to the EnKF for SWE estimation made by site specific LSTMs was within 10 mm (Figure 3, panel e). Similarly, the mean RMSE increase- averaged across sites- compared to the EnKF for snow depth estimation made by site specific LSTMs was equal to 6 cm (Figure 3, panel f). The only exception is the site of FMI-ARC were the LSTM-DA still underperformed compared to the EnKF, although the absolute values of RMSE are 1 order of magnitude lower than the ones on the other sites. The bias analysis (Figure 3, panel g and h) showed that snow depth exhibited a near zero bias, while the LSTM tended to overestimate SWE compared to the EnKF. However, both patterns were consistent in the EnKF and in the S3M open loop.

**Figures 3 & 4:** *The Nash-Sutcliffe coefficient can be used as a score to evaluate the accuracy of the time series in (a)–(d).*

We agree with the reviewer that an additional metric was needed to better evaluate the accuracy of the time series in Figures 3–6. We chose to include the RMSE, as it is a physical quantity that provides a more intuitive understanding of the actual snow values. While we appreciate the comment regarding the use of the Nash-Sutcliffe coefficient, we added the Kling-Gupta Efficiency (KGE) to be more suitable for our purposes, as it better captures both small and large discrepancies in the time series.

Having added the KGE to the Figure we plan to add a few sentences in the results section after line 330:

> When it comes to evaluating the Kling-Gupta Efficiency (KGE) (Gupta et al. 2009), for sites with denser measurements, the values are comparable to those obtained with the EnKF-DA, supporting the observed improvement trend over the open loop simulation. Conversely, in the case of sparse datasets, the lower KGE values highlight the limitations of the LSTM in achieving performances comparable to the EnKF-DA.

After line 354 :

The KGE values, for both dense and sparse datasets, confirm that the memory component primarily acts as a smoother and enhances performance in most scenarios.

**Line 321–324:** *Why is the LSTM trained with outputs (states) from EnKF-DA more sensitive to the sparsity of observation data? Could you explain this here? Including observation data as an input may introduce artificial errors when filling in missing data in the input.*

The LSTM trained with EnKF-DA outputs is more sensitive to the sparsity of observational data because, unlike the EnKF, it lacks the flexibility to dynamically handle missing inputs. In the EnKF framework, the observation operator can be explicitly adjusted to account for the availability or absence of data at each time step, allowing assimilation to proceed even when observations are sparse. In contrast, the LSTM is trained in a supervised manner and requires a complete set of inputs (joint assimilation of SWE and snow depth) at every time step, making it more susceptible to performance degradation under irregular or incomplete observational conditions.

Data sparsity is a well-known challenge in cryospheric science. Therefore, future work will focus on increasing the model's flexibility—exploring alternative neural network architectures, leveraging synthetic data, and explicitly tracking the artificial error such data may introduce.

Despite these limitations, our LSTM provides a lower-bound benchmark for performance, demonstrating the potential for improvement even under high-sparsity conditions—while maintaining the same advantage in computational cost reduction.

**Line 336–337:** *Only Figure 5b shows improvement with memory component, rather than c and d*

We apologize for the absence of coherence between the text and the figure, and we plan to modify the test to adhere to the updated version of the figure 5. Here is the proposed change from line 336 to line 340:

> For datasets characterized by low data sparsity (NGK, KTH, FMI-ARC, RME), incorporating a memory component into the LSTM improved its ability to capture the seasonal dynamics of SWE and snow depth, particularly in accurately representing the timing and magnitude of peak SWE (see Figure 5, panels a and b). However, in some instances (see Figure 5, panels c and d), the memory component did not lead to a significant performance gain. Instead, it primarily acted as a smoother, dampening short-term fluctuations without substantially enhancing predictive accuracy.

**Line 344:** *0.5 m? The reduction shown in figure 6f is not that large.*

We apologize for the mistake in the units of measurements. We will replace it with 0.5 cm

**Line 346:** *These strategies were not mentioned and explained in the method.*
The manual sampling we refer to involves the collection of in-situ SWE data at specific site locations. While we did not describe this process in detail—since it is already well documented in the reference papers for each site—we acknowledge that its brief mention here might be misleading. To clarify this point, we propose revising lines 345–346 as follows:

(e.g., 95%, WFJ and TRG – where the assimilated observations consist of manually measured SWE data, as detailed in the corresponding site references)

**Section 3.3:** *This result does not seem meaningful, as the spatial transferability of all models appears to be poor. Please consider removing it.*

To clarify, our results indicate that the LSTM model trained on the KHT site exhibits a degree of spatial transferability, in some cases even outperforming locally trained models at the test sites. However, we acknowledge the reviewer's concern. Rather than suggesting meaningful spatial transferability at this stage, our intent was to demonstrate the attainment of promising performance with at least one algorithm, setting a lower bound for performances. These findings open new avenues for extending the framework toward a 2D implementation, which could better account for spatial variability and improve generalization across sites.

**Line 370–371:** *Any explanation for this result?*

To our understanding, during wet years, an increase in snow events is observed, which could potentially amplify uncertainties in the model due to cascade effects arising from both precipitation phase partitioning and initial condition uncertainties. Additionally, the formation of multiple snow layers, which may not be fully captured by the physics of our model, further contributes to these complexities. These factors could explain the observed lower performance of the multi-site LSTM during wet years. However, this hypothesis has not been fully tested, and any further explanation would require a comprehensive analysis. Nonetheless, we recognize the need for more detailed discussion, and therefore, we plan to add the following statement after line 371:

> Reduced performances of the Multi-site LSTM simulation on snow water equivalent over wet years may be explained considering the difference in occurrence of snow events during those periods; indeed, in wet years, an increased number of snowfall events may introduce additional complexity and uncertainty, both due to the cascading effects of uncertainties in initial conditions and precipitation phase partitioning (Harder & Pomeroy 2014). Moreover, the formation of several snow layers may not be fully captured by S3M.

**Section 3.4:** *Instead of presenting the spatial transferability of a single model, it might be more meaningful to compare and discuss the site-specific LSTM and the multi-site LSTM. Please refer (this is not my work and no need to cite it.): Kratzert, Frederik, Martin Gauch, Daniel Klotz, and Grey Nearing. "HESS Opinions: Never train a Long Short-Term Memory (LSTM) network on a single basin." Hydrology and Earth System Sciences 28, no. 17 (2024): 4187-4201.*

We appreciate the reviewer's comment and fully acknowledge the growing consensus in the hydrological literature advocating for multi-basin training as a means to achieve more robust and generalizable LSTM streamflow models. However, the purpose of presenting single-site snow results in our study was to explore the lower bounds of snow model performance in a transferability context. Specifically, we aimed to evaluate whether a model trained under such limited conditions could still outperform the open loop and perform comparably to traditional data assimilation techniques.

Among the available datasets, KHT represents the most suitable candidate for this analysis due to its long time series, higher data quality, and relatively large sample size. These attributes make it uniquely valuable for testing spatial transferability and informing the design of future distributed modeling efforts.

It is also important to note that S3M is not a lumped hydrological model, but a spatially distributed (gridded) snow model aimed at simulating snow water equivalent (SWE) across the terrain, rather than its integrated effect on streamflow at a basin outlet. As such, insights from streamflow-focused LSTM models may not transfer directly, given the differing computational units (catchments/basin vs. points/grid cells) and modeling goals.

That said, we agree that there is likely an advantage to training an LSTM-based snow model across multiple spatial locations, enabling the pooling of information in both space and time. That will be a topic of future research, and the transferability experiments in this paper are just a first tentative step in that direction.

To address the reviewer's suggestion and clarify this intention, we will add the following sentence after line 360:

> While recent studies (Kratzert et al. 2024) have strongly advocated for multi-basin training to achieve robust and generalizable LSTM streamflow models, we intentionally present the single-point case here for snow hydrology to establish a performance lower bound for snow spatial transferability—highlighting whether even such a constrained model can outperform the open loop and compare with traditional data assimilation approaches.

**Line 410:** *No results were shown to support this.*

We apologize for the lack of clarity around this point. While we did not present explicit results to support this, our intention was to explore the introduction of soft physical constraints in the cost function as a way to incorporate an inductive bias, as suggested in existing literature (Karniadakis et al. 2021), with the goal of enhancing model generalization. However, this approach did not lead to a notable improvement in model transferability. This suggests that the current level of physics integration may be insufficient, and future efforts should prioritize stronger physics adherence to better support generalization across sites.

we plan to change the sentence at line 410-411 to improve the quality:

> In light of this, we introduced soft physical constraints into the cost function as a way to incorporate an inductive bias. Although this particular approach did not prove effective in significantly enhancing generalization, considerable potential remains in enforcing snow physical constraints in LSTMs (Charbonneau et al. 2024). Further research is needed in this direction to better understand how such constraints can support model generalization and physical consistency.

**Line 415:** *7 sites?*

Here we refer to the site-specific LSTM algorithm ( 1 without and 1 with memory features for all the 7 sites). However the text poorly clarify this aspects so we plan to change the sentence at line 415-416 in:

> All but 2 out of the 14 site-specific LSTM frameworks significantly outperformed the Open Loop, although none outperformed the EnKF.

We plan to modify the manuscript according to these comments :

**Line 254:** Double "the"

**Line 271:** "predictions"

**Line 280:** *The inline formula here should not include 'star,' as 'star' was previously used to represent the LSTM output, not the input from S3M. Please keep consistent.*

**Line 342–348:** *Cite Figure 6 here.*

**References**

Bishop, C. M. & Bishop, H. (2023), *Deep learning: Foundations and concepts*, Springer Nature.

Charbonneau, A., Deck, K. & Schneider, T. (2024), 'A physics-constrained neural differential equation framework for data-driven snowpack simulation', *arXiv preprint arXiv:2412.06819* .

Gupta, H. V., Kling, H., Yilmaz, K. K. & Martinez, G. F. (2009), 'Decomposition of the mean squared error and nse performance criteria: Implications for improving hydrological modelling', *Journal of hydrology* **377**(1-2), 80–91.

Harder, P. & Pomeroy, J. W. (2014), 'Hydrological model uncertainty due to precipitation-phase partitioning methods', *Hydrological Processes* **28**(14), 4311–4327.

Karniadakis, G. E., Kevrekidis, I. G., Lu, L., Perdikaris, P., Wang, S. & Yang, L. (2021), 'Physics-informed machine learning', *Nature Reviews Physics* **3**(6), 422–440.

Kratzert, F., Gauch, M., Klotz, D. & Nearing, G. (2024), 'Hess opinions: Never train a long short-term memory (lstm) network on a single basin', *Hydrology and Earth System Sciences* **28**(17), 4187–4201.

LeCun, Y., Bengio, Y. & Hinton, G. (2015), 'Deep learning', *nature* **521**(7553), 436–444.

Murphy, K. P. (2023), *Probabilistic Machine Learning: Advanced Topics*, MIT Press.
**URL:** *http://probml.github.io/book2*

Prechelt, L. (2002), Early stopping-but when?, *in* 'Neural Networks: Tricks of the trade', Springer, pp. 55–69.

Smith, L. N. (2015), 'Cyclical learning rates for training neural networks. arxiv', *Preprint at https://arxiv. org/abs/1506.01186* .

---

## Author Comment (AC2)

**General comment**

*The paper "Learning to filter: Snow data assimilation using a Long Short-Term Memory network" presents a novel framework for snowpack prediction combining physical-based model and machine learning model. It could be a great fit for the journal. However, there are several aspects of the experimental setup and methodology that would benefit from additional clarification. I encourage the authors to provide more detailed descriptions of their experiments to enhance the transparency and reproducibility of the study. Please see my comments below.*

We thank the reviewer for the positive evaluation of our manuscript and we acknowledge the need for additional clarification needed to sustain transparency and reproducibility of our study. We plan to improve the quality thanks to the useful feedback received. Here below is a list of answers to specific comment and planned changes.

**Specific Comments**

*The overall data samples are limited (both years and sites), compared to https://doi.org/10.1175/JHM-D-22-0220.1 Could the authors comment on this issue?*

We acknowledge the reviewer's concern. Our aim was to explore the application of pilot point approach across different locations in the Northern Hemisphere that would be fit for operational and quasi-real time applications. Despite working with a relatively limited dataset, the results are still promising and provide a solid foundation for future developments, especially in a 2D spatially distributed modeling case. Nevertheless, we acknowledge the importance to point out this difference in our work, hence we plan to add a sentence in the introduction at line and one in the discussion at line . Here the two sentences
Introduction: at line 83 after "based on topographic features.":

> As more recent exception of combining deep learning and snow data assimilation, Song et al. (2024) developed an LSTM-based framework to assimilate lagged observations of SWE or satellite-derived snow cover fraction (SCF) over the western U.S., aiming to improve seasonal snow predictions. While their approach further consolidates the potential of deep learning for data assimilation in snow hydrology, it relied on a relatively simple assimilation setup, dealing with long lagged time step rather than a consequential and quasi real time approach.

Discussion: at line 407 after "used in training".:

> Although using limited datasets in both temporal and spatial coverage compared to recent studies pursuing a similar effort (Song et al. 2024), our approach proved to be effective in speeding up traditional data assimilation techniques while maintaining comparable performance; additionally , our framework, designed to test the operational viability of a quasi–real-time pilot point method, still proved the feasibility of an alternative use of LSTM algorithm without loosing in performances. The encouraging results provide a foundation for extending this framework to broader, more diverse networks in future research.

*What is the temporal frequency of S3M? Is it 1 hour (**Line 121**)?*

The temporal frequency used in this study is 1 hour, but S3M can be run with different time steps.

*What is the input time window size for the LSTM model? If my understanding is correct, only one timestep of meteorological forcings are used as input, based on Fig. 2 and Line 269. This is not a typical use of the LSTM model if multi-time steps are not involved. The architecture of the LSTM model also requires more details (e.g., hidden layers, hidden units).*

We appreciate the reviewer's insightful comment. Indeed, during the operational testing phase, the LSTM model is provided with only one timestep of meteorological forcings as input. However, this design is intentional, aimed at simulating real-time forecasting conditions, where only the current timestep of meteorological data is available for prediction. During the training phase, the model is trained with multi-time-step sequences to learn temporal dependencies, consistent with traditional LSTM approaches. Therefore, although the operational phase uses only one timestep of input, the model is trained with multi-timestep data to capture temporal dynamics over time. Concerning the architecture of the LSTM model , we plan to add a clarifying section as mentioned in the answer to RC1

*Related to the previous comment, please clarify the "memory components" of the LSTM model. By design, the previous time series should be used as inputs to the LSTM model. What is the model without these "memory components"? If this is beneficial, do the authors consider incorporating more previous timesteps?*

The "memory components" refer to additional sets of features provided to the LSTM during both the training and testing phases. These components include:

- The meteorological forcing variables at each timestep (i.e., timestep $k$), and

- The state of the system at the previous timestep (i.e., at timestep $k - 1$).

Regarding the reviewer's suggestion of incorporating more previous timesteps: we acknowledge that this could improve the model's predictive capability, particularly during the operational phase. We will consider exploring the use of longer input sequences of previous timesteps in future work, contingent upon operational constraints.

*Loss function. As noted in **Line 246**, the output of negative SWE is forced back to zero, why is the regularization term still necessary in **Line 260**? Is the hard cut at zero only applied after training the model?*

We acknowledge that the current structure of the paragraph may lead to a misleading interpretation of the procedure. Specifically, the hard cut to zero was applied after the LSTM prediction, and therefore after the regularization term was used. This step was introduced to preserve the intermittency of snow quantities while also helping the LSTM network learn to detect the onset of a snowpack with new snowfall on bare ground. To enhance clarity, we plan to move the sentence currently at lines 246–247 to follow line 265.

*Multisite LSTM. Do the authors consider the use of site-specific information as inputs (e.g., lat-lon, slope https://agupubs.onlinelibrary.wiley.com/doi/full/10.1029/2023WR035009; https://agupubs.onlinelibrary.wiley.com/doi/10.1029/2021WR031033)*

We did not used elevation and coordinates of the site, but rather decide to decrease the number of input as to match those used by the EnKF in a non-distributed modelling. However, we plan to develop a 2D spatially distributed snow modeling framework of our LSTM EnKF emulator using more basins and data within each basin to leverage such info.

***Line 326***. *"Reduce" RMSE by "-25" seems to increase RMSE for me. Please consider rephrasing it.*

We apologize for the lack of clarity and we will improve the revised manuscript. We plan to modify the sentence at line 326 as follows:

> indeed the LSTM resulted in a reduction of 25 mm in RMSE for SWE, while the EnKF achieved a larger reduction of 31 mm.

**Figure 5**. *Why is the RMSE for "open loop" not shown here?*

We have modified the figures 5 and 6 to show also the RMSE of the open loop.

*There are some caption inconsistencies. Please take time and revise them (e.g., the capital letters in Figure 8 caption)*

We thank the reviewer for pointing out these inconsistencies, we will revise and correct the caption.

**Figure 8**. *Is there any particular reason to assess the performance based on different water year types? A similar and consistent RMSE as previous experiments would be helpful.*

The rationale behind evaluating performance across different water year types was to test the model's robustness under varying hydrological conditions, such as dry, normal, and wet years. This, according to our opinion and based on well established procedures in hydrology(Osuch et al. 2015), allows us to better understand how the model performs beyond average conditions, particularly in more challenging or extreme scenarios.

**References**

Osuch, M., Romanowicz, R. J. & Booij, M. J. (2015), 'The influence of parametric uncertainty on the relationships between hbv model parameters and climatic characteristics', *Hydrological Sciences Journal* **60**(7-8), 1299–1316.

Song, Y., Tsai, W.-P., Gluck, J., Rhoades, A., Zarzycki, C., McCrary, R., Lawson, K. & Shen, C. (2024), 'Lstm-based data integration to improve snow water equivalent prediction and diagnose error sources', *Journal of Hydrometeorology* **25**(1), 223–237.

---

## Referee Report (RR1)

**General comments**

The topic of the manuscript is interesting and relevant for the readers of Cryosphere, as well as for the larger international hydrology community. It is also in line with the actual popularity of machine learning in hydrology (and everywhere, really!). If I understand the situation correctly, the manuscript has already been reviewed by two anonymous referrees, who might not be available anymore, or the editor wanted a third opinion. Considering this, I have tried my best to prioritise verifying that the authors have addressed all the comments made by the two initial referees and I will refrain from starting the review process all over again. I did, however, notice a few minor points that were not raised during the initial review that I think would be wort addressing. Overall, I would consider this a minor revision.

**Specific comments**

1. Introduction: too much emphasis on mountains

The introduction of the manuscript start with emphasising the importance of snow in mountain regions. As someone who lives in a non-mountainous area where snow plays a very important role in the hydrological cycle, I tend to find this a little bit annoying. The data assimilation method you propose is presented as a point-based method, at least for now. One of the challenges of modelling (or assimilating) snow in mountainous regions is the high spatial variability, and this is something that the current version of your method is not addressing. However, your method is still interesting an useful, in general, for snow dominated areas. All of this to say: I think it would make more sense to start the introduction from a more general point of view, and not « mountains » specifically. For instance, you could just start the introduction with « reliable estimates of snow water equivalent (SWE) and snow depth in snow-dominated environments are essential (...) », explain why, and then maybe mention the specific case of mountains.

2. Data quality: how do you mesure it?

Section 2.1 mentions that you used «(...) high-quality, pre-processed datasets from long-term, internationally acknowledged snow research stations across the northern hemisphere (...) ». Can you specify how data quality was mesured and provide some details?

3. Data quality: how robust is the proposed method?

Maybe this comment is more for the discussion, and it also depends on your reply to my previous question, but I think it would be interesting to reflect on how robust you think the method will be to lower quality data? Operationally, one would want to apply your method using data of slightly lower quality, if that is what is available to them. How will this affect the results? Would the method be able to « ignore » a portion of the data if the quality is too low?

4. Reviewer 1's comment about section 3.4 (spatial transferability vs multisite training)

The original Reviewer 1 commented that « instead of presenting the spatial transferability of a single model, it might be more meaningful to compare and discuss the site-specific LSTM and the multi-site LSTM ».

I completely agree with them, and I don't think the authors have addressed that comment satisfactorily. I understand your concern and interest with exploring the lower bounds of snow model performance, etc., but this comparison between single-site and multi-site training is one of the most relevant aspect of the paper, considering the current literature in hydrology and LSTMs. In that sense, I really don't think that the addition you made after line 360 is sufficient. I would strongly encourage you to provide a more thorough comparison, as it would be of very high interest to the scientific community.

On the same topic, I found Figure 8 extrêmely difficult to interpret, even just on its own, and impossible to compare with Figures 3 to 6. Please provide a figure that will allow the readers to directly compare the results of single site vs multi-site training.

Theoretically, it seems like a good idea to train the LSTMs on multiple sites, as it provides them with more data, and more diversity. Is it the case in practice for data assimilation? This should be discussed.

Also regarding Figure 8: please write « probability density function » instead of the acronym « PDF » or, even better, remove the title above the figure altogether, keeping only the caption below.

5. Reviewer 2's general comment about the lack of details in the description of your experiments

I agree with this comment and I acknowledge the efforts that you already made to improve this. However, I think some important details are still missing or too vague:

- What is the programming language that was used to build the LSTMs? Any specific toolboxes or libraries?
- In section 2.4.4: what is the length of the lookback window? Related to that, Rev 2 asked to « please clarify the memory component (...) ». Normally, LSTMs do not need to be provided with data from previous timesteps, because of the lookback window (sometimes called lookback period, or just lookback). This is an important advantage of LSTMs compared to multilayer perceptrons, for instance, which have no memory. As their name indicates, LSTMs were designed specifically to have a memory. It is difficult to understand why you are adding a supplementary memory component. Including input variables form previous time steps is something we would typically do with a MLP, because of their lack of memory. Therefore, in addition to specifying the length of the lookback window, can you explain how adding more time steps for input variables is not redundant with the lookback window?
- Rev 2 had a specific comment about « Furthermore, any LSTM prediction that fell below zero was forced back to the zero, effectively managing intermittent nature of snow data. ». You modified the sentence and moved it to lines 271-272, but in my opinion this is still not clear. How exactly was this « forcing back to zero » performed? I guess that you determined specific dates for each one of the 7 study sites where there should not be any snow, and then for each year, between those dates you replaced the LSTM prediction by zeros? If this is the case, can you please explain how the dates were determined for each site?
- 6. A point for the discussion: how to move from the current model to a spatialized version.

As you mention, S3M is a distributed model. In the discussion, you briefly mention that « 'These results open a window of opportunity for spatially distributed deep data assimilation; hence future work should focus on testing such a spatio-temporal water configuration. » Could you please expand on what would need to be modified in the proposed data assimilated method in order for it to be applied in a spatially distributed way?

**Minor comments, typos**

- Figures A1 and A2: the text on those figures is much too small. Please make it bigger
- Line 1999 there is a space missing here « by Reichle et al. (2007), Lanoy et al. (2010) »
- Line 285: there is a missing space before the new part that was added in blue.
- Line 299: there is a missing comma before « cycling between »
- Figures 3-4-5-6, replace « pan » by « panel »
- Beginning of section 3.1, line 386: remove one of the « whiles » in « while while »

---

## Referee Report (RR2)

**Second revision of « Learning to filter: Snow data assimilation using a Long Short-Term Memory network » by Blandini et al.**

The authors have responded to all my questions and comments satisfactorily. I have no further comments, and I recommend the publication of the manuscript in its current form.

---

## Author Response (AR2)

**Reply to RC.1**

**General comments**

The authors of "Learning to filter: Snow data assimilation using a Long Short-Term Memory network" have made efforts to address previous comments. Please see my comment below for further clarification and improvement.

We thank the reviewer for the time dedicated to the evaluation of our manuscript to further improve its quality and to make it better understandable to a broader audience. We modified the manuscript following reviewer's comments, as we detail below.

**Specific comments**

1. Please consider adding longitude lines for the European sites, and one more latitude line for both European and Asian sites in Figure 1.

Comment addressed.

2. If my understanding is correct, the LSTM model—at least in the operational testing configuration—uses only a single previous timestep as input. The main difference compared to a basic model appears to be that the LSTM's memory component includes meteorological forcings from the previous timestep (as shown in Equation 4). This raises the question: is the use of an LSTM necessary in this case? Would a simpler ANN model suffice? Additionally, the authors state that this setting is to mimic the real forecast setting, but it is unclear why this is the case. Typically, historical meteorological data should be accessible for forecasting applications. Could the authors clarify this point? Finally, the response mentions that the training phase uses multi-time-step input sequences. What is the specific timestep length used during training? Does this imply that the model is trained using a different temporal input structure than what is used during testing? If so, a more explicit explanation of this mismatch would be helpful.

As correctly understood by the reviewer, the operational testing configuration of the LSTM model indeed uses only the previous single timestep as input. However, contrary to a basic Artificial Neural Network, the LSTM can retain and implicitly utilize information from a longer sequence of past time steps through its internal cell states and hidden states previously trained. This ability to maintain a "memory" of historical conditions, even when only a single previous timestep is given as input, is why we choose to use a LSTM even while relying on this alternative single-step testing setup. Additionally, during the training phase, the model was trained using multi-time-step input sequences. Specifically, approximately 10 years of hourly data records for each site were used during training. Acknowledging that the different approach between training and testing may not be satisfactory explained, we added an explanatory comment on line 304 after "... observations and model predictions.:

"It is important to stress that, while the training phase was performed in the conventional way of training neural networks -meaning multiple timestep as input

to obtain a sequence of outputs - the operational testing phase was performed giving to the LSTM trained models only one timestep at a time, to be coupled with the forward step of the cryospheric model."

Finally, we agree with the reviewer that historical meteorological data is accessible for model development and training, in a true operational forecast scenario; however, we would like to point out that the framework we built was to couple our LSTM as an online sequential data assimilation step (i.e., so-called filtering) to re-initialize our cryospheric model, which relies only on the most current observations and its internal learned dynamics during each computational step to project snow conditions forward. This argument is particularly important in terms of reducing the amount of storage needed to run a model operationally for forecasting purposes. This is why we chose to only rely on the most recent timestep in testing.

3. Figure 5. Please check the colors in the legend. It seems to me the color for LSTM-DA and LSTM-DA with memory in the legend is too light and do not match with the figure.

Comment addressed.

**Reply to RC.2**

**General comments**

I think the authors have done good revision work and improved the quality of this manuscript. My concerns have been addressed or explained.

We appreciate the help of the reviewer in further improving the quality of the manuscript and we will perform an in-depth grammar and typo check before submitting the new version.

**Specific comments**

I only have two minor comments left:

1. The colors of the result figures are difficult to distinguish between the different models. Please further improve the clarity of these figures. Figure A1: The legend is hard to read.

Comment addressed.

2. Please correct the grammar and typo errors throughout the manuscript.

For example:

Line 403: "KGE" should not have a unit. Line 498: "without loss in performance."

There are more besides these two examples.

Comment addressed.

**Reply to RC.3**

**General comments**

The topic of the manuscript is interesting and relevant for the readers of Cryosphere, as well as for the larger international hydrology community. It is also in line with the actual popularity of machine learning in hydrology (and everywhere, really!). If I understand the situation correctly, the manuscript has already been reviewed by two anonymous referees, who might not be available anymore, or the editor wanted a third opinion. Considering this, I have tried my best to prioritise verifying that the authors have addressed all the comments made by the two initial referees and I will refrain from starting the review process all over again. I did, however, notice a few minor points that were not raised during the initial review that I think would be wort addressing. Overall, I would consider this a minor revision

We appreciate the reviewer comment on the impact of our paper on the Cryosphere community and we thank them for the further effort made to help us improve our manuscript. We provide here more in-detail answers to reviewer's comments and modified the manuscript following the reviewer's suggestions.

**Specific comments**

1. Introduction: too much emphasis on mountains- The introduction of the manuscript start with emphasising the importance of snow in mountain regions. As someone who lives in a non-mountainous area where snow plays a very important role in the hydrological cycle, I tend to nd this a little bit annoying. The data assimilation method you propose is presented as a point-based method, at least for now. One of the challenges of modelling (or assimilating) snow in mountainous regions is the high spatial variability, and this is something that the current version of your method is not addressing. However, your method is still interesting an useful, in general, for snow dominated areas. All of this to say: I think it would make more sense to start the introduction from a more general point of view, and not "mountains" specially. For instance, you could just start the introduction with "reliable estimates of snow water equivalent (SWE) and snow depth in snow-dominated environments are essential (...) ",explain why, and then maybe mention the specific case of mountains.

We understand the reviewer's concern. We addressed this comment by changing the introduction as follows:

"When studying the hydrological cycle, one cannot underestimate the key role played by snow (Pagano & Sorooshian 2002); indeed, for snow-dominated catchments, today's snow is tomorrow's water. Information on the state and distribution of snow cover provides helpful information to characterize seasonal water storage (Zakeri et al. 2024), seasonal to annual water availability (Metref et al. 2023), and several cascading socio-hydrologic implications (Avanzi et al. 2024).

Especially in cold regions, which are heavily affected by climate change (Hock et al. 2019), the snowpack often functions as the primary source of streamflow, particularly during spring and summer (Bales et al. 2006). Moreover, considering the high spatial variability in these regions, the scientific community agrees on the needs of reliable estimates of Snow Water Equivalent (SWE) and snow depth in snow-dominated environments, which are essential for effective and timely management of water resources (Hartman et al. 1995)."

2. Data quality: how do you measure it? Section 2.1 mentions that you used '(...) high-quality, pre-processed datasets from long-term, internationally acknowledged snow research stations across the northern hemisphere (...) '. Can you specify how data quality was measured and provide some details?

To assess the quality of the dataset used in our study, we relied on the quality control procedures and documentation provided by the original data providers, who typically apply standardized protocols for data collection, sensor calibration, and outlier removal. These datasets are widely used and cited in the scientific community, and their quality assurance practices are well established. While we did not perform an independent quality assessment, we selected stations with a long history of data availability and peer-reviewed documentation to ensure the reliability of the input used in our analysis.

3. Data quality: how robust is the proposed method? Maybe this comment is more for the discussion, and it also depends on your reply to my previous question, but I think it would be interesting to reflect on how robust you think the method will be to lower quality data? Operationally, one would want to apply your method using data of slightly lower quality, if that is what is available to them. How will this affect the results? Would the method be able to 'ignore' a portion of the data if the quality is too low?

We appreciate this timely and relevant question, which indeed opens a broader discussion around the applicability of artificial intelligence-based (AI) methods in real-world operational contexts. It is well known that AI and, in particular, deep learning techniques are highly sensitive to data quality and quantity. The performance of these models tends to degrade when trained or applied to noisy, incomplete, or low-resolution data.

However, this limitation can often be addressed through robust data pre-processing and augmentation strategies. Indeed, a significant portion of time in AI workflows is devoted to improving data quality before model training. Moreover, there is growing interest in hybrid approaches that combine traditional statistical techniques with deep learning, offering a potential pathway to enhance robustness in low-quality data settings.

For instance, recent work by Gauch et al. (2025) provides valuable insights into handling missing data in operational environments, suggesting imputation and correction methods that could be integrated into AI pipelines. Additionally, On the AI side, generative models (e.g., GANs or diffusion-based models) have shown promise in enriching and recovering incomplete datasets, as illustrated by Dhoni (2023).

In order to add a discussion of this timely topic to our paper, we have added a sentence after line 557:

While our results are based on high-quality forcing and observational datasets, we acknowledge that operational applications may involve lower-quality inputs. In such cases, pre-processing strategies (e.g., bias correction, gap-filling) and hybrid DA-AI frameworks could help mitigate performance loss, with the potential to selectively down-weight unreliable inputs rather than propagating their errors through the model. Recent work by Gauch et al. (2025) demonstrates the effectiveness of imputation and correction methods for handling missing or degraded data in operational environments, while generative models such as those explored by Dhoni (2023) offer promising avenues for enriching and augmenting incomplete datasets.

4. Reviewer 1's comment about section 3.4 (spatial transferability vs multisite training) The original Reviewer 1 commented that 'instead of presenting the spatial transferability of a single model, it might be more meaningful to compare and discuss the site-specific LSTM and the multi-site LSTM. I completely agree with them, and I don't think the authors have addressed that comment satisfactorily. I understand your concern and interest with exploring the lower bounds of snow model performance, etc., but this comparison between single-site and multi-site training is one of the most relevant aspect of the paper, considering the current literature in hydrology and LSTMs. In that sense, I really don't think that the addition you made after line 360 is sufficient. I would strongly encourage you to provide a more thorough comparison, as it would be of very high interest to the scientific community. On the same topic, I found Figure 8 extremely difficult to interpret, even just on its own, and impossible to compare with Figures 3 to 6. Please provide a figure that will allow the readers to directly compare the results of single site vs multi-site training. Theoretically, it seems like a good idea to train the LSTMs on multiple sites, as it provides them with more data, and more diversity. Is it the case in practice for data assimilation? This should be discussed. Also regarding Figure 8: please write 'probability density function' instead of the acronym 'PDF' or, even better, remove the title above the figure altogether, keeping only the caption below.

We acknowledge the point of the reviewer and we now provide a comparison between the best site-specific LSTM and the multiple site LSTM. We added a new section in the Results:

Comparing the multi-site LSTM DA with the site-specific LSTM DA trained over KHT, results show comparable performance for SWE, with neither approach consistently outperforming the other(see Fig. 9). In some cases, the site-specific model achieves lower errors, while in others the multi-site model performs equally well or slightly better. For snow depth, however, the multi-site LSTM DA tends to outperform the site-specific LSTM DA across most sites, although the improvements are generally modest (e.g. see Fig. 9 pannel d)

Then, in the discussion section, after line 544, we added this paragraph:

Additionally, considering a comparison between two approach, neither the site-specific nor the multi-site LSTM-DA consistently outperforms the other. While

multi-site training is theoretically expected to improve generalization by exposing the model to a broader range of conditions, this benefit is not clearly observed for SWE. A likely reason could be an uneven representation of sites, combined with variability in snowpack properties, meteorological drivers, and measurement methods, which may bias the model and introduce noise, leading to underfitting. Snow density also plays a crucial role; SWE is defined as  $W = d\rho$ , where W is SWE  $[kg/m^2]$ , d is snow depth [m], and  $\rho$  is bulk snow density  $[kg/m^3]$ . A site-specific model such as KHT may implicitly capture a representative density evolution that transfers well across sites, whereas a multi-site model must attempt to generalize density dynamics across all environments, often with less accuracy. Overall, the multi-site LSTM-DA and the EnKF-DA perform similarly, with the latter only marginally better. This is encouraging, as it highlights the potential of the multi-site LSTM-DA to achieve comparable performance while substantially reducing the computational cost associated with ensemble-based methods.

Additionally, We modified the title on figure 8 (Now figure 9) accordingly to reviewer's suggestion.

- 5. Reviewer 2's general comment about the lack of details in the description of your experiments. I agree with this comment and I acknowledge the efforts that you already made to improve this. However, I think some important details are still missing or too vague:
- What is the programming language that was used to build the LSTMs? Any specific toolboxes or libraries?

To address this comment we added a sentence after line 282:

To develop the LSTM algorithm, we used Python 3.9.21 programming language and the open source libraries Keras v.2.10.0 (Chollet et al. 2015) and Scikit-learn v.1.1.1 (Pedregosa et al. 2011).

-In section 2.4.4: what is the length of the lookback window? Related to that, Rev 2 asked to 'please clarify the memory component (...) '. Normally, LSTMs do not need to be provided with data from previous timesteps, because of the lookback window (sometimes called lookback period, or just lookback). This is an important advantage of LSTMs compared to multilayer perceptrons, for instance, which have no memory. As their name indicates, LSTMs were designed specifically to have a memory. It is difficult to understand why you are adding a supplementary memory component. Including input variables form previous time steps is something we would typically do with a MLP, because of their lack of memory. Therefore, in addition to specifying the length of the lookback window, can you explain how adding more time steps for input variables is not redundant with the lookback window?

The lookback in the training phase is on average 10 years of hourly data (24x365x10), following the comments of Liu et al. (2015) who raised the concern that for long sequences the important information from the beginning of the sequence has to be dragged through the

whole sequence. Then during the testing the lookback is ether 1 or 2 hours (in the memory component configuration). So in that sense the memory component configuration name is used to stress a "longer" memory call rather than a shorter memory component usage. We acknowledge this may still be not so clear so we modified the sentences at line 311 as follows:

"The second test configuration introduced an additional feature component to call back on the use of the "long" memory component of the LSTM during the operational test phase."

- Rev 2 had a specific comment about 'Furthermore, any LSTM prediction that fell below zero was forced back to the zero, effectively managing intermittent nature of snow data. '. You modified the sentence and moved it to lines 271-272, but in my opinion this is still not clear. How exactly was this 'forcing back to zero' performed? I guess that you determined specific dates for each one of the 7 study sites where there should not be any snow, and then for each year, between those dates you replaced the LSTM prediction by zeros? If this is the case, can you please explain how the dates were determined for each site?

The 'forcing back to zero' was a direct post-processing step applied to the LSTMs output. Any negative predictions for Snow Water Equivalent or snow depth was forced back to zero by applying the ramp function. This was a simple, per-timestep constraint on the model's output, not a process based on predefined snow-free periods or specific dates for each study site.

6. A point for the discussion: how to move from the current model to a spatialized version. As you mention, S3M is a distributed model. In the discussion, you briefly mention that 'These results open a window of opportunity for spatially distributed deep data assimilation; hence future work should focus on testing such a spatio-temporal water configuration.' Could you please expand on what would need to be modified in the proposed data assimilated method in order for it to be applied in a spatially distributed way?

We thank the reviewer for the opportunity to further expand on this point and improve the clarity of our manuscript. We have added a comment after line 570 after the sentence -"LSTM robustness during dry and average water years further underscores the generalization capacity of such a framework.".

"Using the LSTM as an emulator of the ensemble Kalman Filter allows us to significantly reduce the computational cost of ensemble-based data assimilation. This is particularly advantageous in a spatially distributed configuration, where running a full ensemble over large domains could otherwise be prohibitively expensive. In our current setup, ensembles are required only during the LSTM training phase, not at inference time — resulting in a more efficient approach for operational use.

Preliminary tests with multi-site LSTM configurations have shown promising results: a single LSTM model trained on data from multiple locations can generalize well to other sites. Building on this idea, we envision extending the application of LSTMs from single-point setups to multiple representative points within a catchment. This spatially sparse assimilation could then be combined with an interpolation or spatial mapping strategy to propagate the correction across the entire domain.

Such an approach would provide a practical compromise between the need for spatially distributed corrections and the computational limitations of full-domain deep data assimilation. This transition from point-based to distributed correction—leveraging spatial generalization and interpolation—will be a key focus of our future work."

**Minor comments, typos**

- Figures A1 and A2: the text on those figures is much too small. Please make it bigger
- Line 1999 there is a space missing here 'by Reichle et al. (2007), Lanoy et al. (2010) '
- Line 285: there is a missing space before the new part that was added in blue.
- Line 299: there is a missing comma before 'cycling between'
- Figures 3-4-5-6, replace 'pan' by 'panel'
- Beginning of section 3.1, line 386: remove one of the 'whiles' in 'while while'

Comments addressed.

**References**

- Avanzi, F., Munerol, F., Milelli, M., Gabellani, S., Massari, C., Girotto, M., Cremonese, E., Galvagno, M., Bruno, G., Morra di Cella, U. et al. (2024), 'Winter snow deficit was a harbinger of summer 2022 socio-hydrologic drought in the po basin, italy', *Communications Earth & Environment* 5(1), 64.
- Bales, R. C., Molotch, N. P., Painter, T. H., Dettinger, M. D., Rice, R. & Dozier, J. (2006), 'Mountain hydrology of the western united states', Water Resources Research 42(8).
- Chollet, F. et al. (2015), 'Keras', https://keras.io.
- Dhoni, P. S. (2023), 'Enhancing data quality through generative ai: An empirical study with data', Authorea Preprints.
- Gauch, M., Kratzert, F., Klotz, D., Nearing, G., Cohen, D. & Gilon, O. (2025), 'How to deal with missing input data', *EGUsphere* **2025**, 1–21.
  - URL: https://egusphere.copernicus.org/preprints/2025/egusphere-2025-1224/
- Hartman, R. K., Rost, A. A. & Anderson, D. M. (1995), 'Operational processing of multi-source snow data', *Proceedings of the Western Snow Conference* **147**, 151.
- Hock, R., Rasul, G., Adler, C., Cáceres, B., Gruber, S., Hirabayashi, Y., Jackson, M., Kääb, A., Kang, S., Kutuzov, S. et al. (2019), 'High mountain areas supplementary material', *IPCC Special Report on the Ocean and Cryosphere in a Changing Climate*.
- Liu, P., Qiu, X., Chen, X., Wu, S. & Huang, X.-J. (2015), Multi-timescale long short-term memory neural network for modelling sentences and documents, *in* 'Proceedings of the 2015 conference on empirical methods in natural language processing', pp. 2326–2335.
- Metref, S., Cosme, E., Le Lay, M. & Gailhard, J. (2023), 'Snow data assimilation for seasonal streamflow supply prediction in mountainous basins', *Hydrology and Earth System Sciences* **27**(12), 2283–2299.
  - **URL:** https://hess.copernicus.org/articles/27/2283/2023/
- Pagano, T. & Sorooshian, S. (2002), 'Hydrologic cycle', University of Arizona. Tucson. AZ. USA.
- Pedregosa, F., Varoquaux, G., Gramfort, A., Michel, V., Thirion, B., Grisel, O., Blondel, M., Prettenhofer, P., Weiss, R., Dubourg, V., Vanderplas, J., Passos, A., Cournapeau, D., Brucher, M., Perrot, M. & Duchesnay, E. (2011), 'Scikit-learn: Machine learning in Python', Journal of Machine Learning Research 12, 2825–2830.
- Zakeri, F., Mariethoz, G. & Girotto, M. (2024), 'High-resolution snow water equivalent estimation: A data-driven method for localized downscaling of climate data', *EGUsphere* **2024**, 1–30.